# Translational fidelity and longevity are genetically linked

Boyang Zheng[1,2,8], Weijie Zhang[1,2,8], Gongwang Yu[3,8], Wenjun Shi[2], Shuyun Deng[2], Xiaoyi Zhang[1,2], Jingyu Chen[2], Ziwei Zhou[1,2], Yuyan Shan[1,2], Wanting Wu[4], Erping Long ®[5], Xiaoshu Chen ®[1,4,6,7] ✉ & Jian-Rong Yang ®[1,2,6,7] ✉

Aging is a series of adverse changes over time that increases mortality risk. Several hypotheses have been proposed to explain aging, including Leslie Orgel's Error-Catastrophe Theory, which asserts that translation errors erode the translational machinery, become self-amplifying, and eventually lead to death. Evidence for the theory is scarce, especially regarding intra-specific fidelity-longevity correlations. Here, we demonstrate that the correlation can be hidden by the constrained evolution of translational fidelity, but remains detectable in long-lived samples. Measuring the lifespan and translational fidelity of a panel of BY × RM yeast recombinant haploid progenies, we validate the fidelity-longevity correlation. QTL analyses reveal that both fidelity and longevity are most strongly associated with a locus encoding vacuolar protein sorting-associated protein 70(VPS70). Replacing *VPS70* in BY by its RM allele reduces translation error by ~8.0% and extends lifespan by ~8.9% through a vacuole-dependent mechanism. Our results support the impact of translational fidelity on intra-specific longevity variation.

Aging is the progressive decline in the physiological function and increase of vulnerability to death observed in most living organisms[1,2]. A number of theories have been proposed to explain aging from various perspectives, including proximate causes such as the somatic mutation theory[3], the telomere theory[4], the free radical damage theory[5], the autoimmune theory[6], and ultimate causes/evolutionary origins such as the antagonistic pleiotropy theory[7] and the mutation accumulation theory[8]. One of the most influential of them is the Error-Catastrophe Theory of Aging[9], first proposed by Leslie Orgel in 1963. According to this theory, errors that inevitably occur during mRNA translation will, sooner or later, happen to the proteins involved in the molecular machinery of translation. Consequently, translational

fidelity will be reduced, resulting in a vicious circle towards even more errors, thus the decline of physiological function and eventually the death of the organism[9,10].

Based on the Error-Catastrophe Theory of Aging, there are three major directions to test the role of translation error in aging. FIRST, the theory predicts that aged cells will produce more erroneous proteins than young cells. However, this has not been supported by a large body of experimental results from a variety of organisms, which have used two-dimensional gel electrophoresis[11–15], viral probes[16–20], or other in vitro read-outs[21–24] to estimate translation error rates. Nevertheless, it is possible that these negative results are due to the limited resolution/accuracy of the utilized techniques, especially for translation errors in

[1]Advanced Medical Technology Center, The First Affiliated Hospital, Zhongshan School of Medicine, Sun Yat-sen University, Guangzhou, China. [2]Department of Genetics and Biomedical Informatics, Zhongshan School of Medicine, Sun Yat-sen University, Guangzhou, China. [3]The Affiliated Dongguan Songshan Lake Central Hospital, Guangdong Medical University, Dongguan, China. [4]Department of Immunology and Microbiology, Zhongshan School of Medicine, Sun Yat-sen University, Guangzhou, China. [5]Institute of Basic Medical Sciences, Chinese Academy of Medical Sciences and Peking Union Medical College, Beijing, China. [6]Key Laboratory of Tropical Disease Control, Ministry of Education, Sun Yat-sen University, Guangzhou, China. [7]Guangdong Provincial Highly Pathogenic Microorganism Science Data Center, Guangzhou, China. [8]These authors contributed equally: Boyang Zheng, Weijie Zhang, Gongwang Yu. ✉e-mail: chenxshu3@mail.sysu.edu.cn; yangjianrong@mail.sysu.edu.cn

low-abundance proteins[25,26], and/or the inability of in vitro assays to represent in vivo changes of translational fidelity[27]. SECOND, the theory predicts a correlation between longevity and translational fidelity, which has been demonstrated by comparisons across species[27]. Nevertheless, despite being more relevant to in vivo natural situations, such comparative analyses may be confounded by the vast genetic differences between species. A THIRD prediction of the theory is that longevity should change accordingly as translational fidelity is manipulated. Early experiments in this direction, in which streptomycin was used to enhance translation error rates, had largely negative results[28,29]. But more recently, some positive results have been obtained when paromomycin or mutant ribosomal proteins are used to increase translation error rate[30–32]. While the use of specific antibiotics or artificial mutations might not reflect the natural conditions, these findings demonstrated that increased translational fidelity can indeed enhance longevity, and prompted renewed interest in the theory. Overall, previous research has neither refuted nor proven the role of translation error in aging, especially regarding whether it underlies the intra-specific longevity variation such as observed in humans[33].

In this context, recent technological advances have enabled more sensitive and comprehensive detection of translation error. For example, luciferase reporter systems can detect changes in translational fidelity due to tRNA availability[25,34] and mRNA secondary structure[35]. More recently, mass spectrometry-based detection of amino acid misincorporation has enabled systematic identification of translation errors at the genome scale[36]. Due to these improved experimental methods, changes in translational fidelity have been revealed as highly pleiotropic. For example, higher translational fidelity reduces the amount of mistranslated erroneous proteins and cellular toxicity associated with translation error-induced protein misfolding[35,37–39], whereas lower translational fidelity increases survival under stressful environments[40] and facilitates evolvability[41,42]. As a result of this pleiotropic constraint (and potentially also the drift barrier[43–45]), translational fidelity is expected to exhibit a narrow range of variation[46–49], which may have hindered the detection of the fidelity-longevity correlation.

In this study, we aimed to test translation error's association with longevity and its underlying genetic basis in light of the above considerations. By a theoretical derivation based on Orgel's original model, we demonstrated that a genetic correlation between translational fidelity and longevity could have been obscured by the limited variation in translational fidelity, but can be recovered by focusing on long-lived samples. To empirically test the fidelity-longevity correlation, we measured the translation error rate and chronological lifespan for a panel of 235 strains from *Saccharomyces cerevisiae* BY × RM recombinant haploid progeny (segregants). Consistent with our model, we found significant fidelity-longevity correlation when only the long-lived but not all segregants were analyzed. Genome-wide quantitative trait loci (QTL) mapping based on this subpopulation identified two and one loci significantly linked to translational fidelity and longevity, respectively. Intriguingly, the most significant loci for both traits overlap at chrX:641,753-669,427. Further experiments on individual genes in this region demonstrated that both translational fidelity and longevity were significantly increased by replacing the gene *VPS70* (Vacuolar protein sorting-associated protein 70) in BY with its RM version, an effect that could be mitigated by an inhibitor of vacuolar function. These results collectively demonstrated the genetic basis for the correlation between translation error and aging, which strongly support the role of translational fidelity in intra-specific longevity variations.

## Results

### The translational fidelity-longevity correlation concealed by the limited variation of translation error rate

To examine the intra-specific correlation between translational fidelity and longevity, we revisited Orgel's mathematical model for translation

error propagation[10,50] (Fig. 1a). With $e_t$ denoting the aggregate translation error rate at time $t$, and $E$ denoting the baseline translation error rate, Orgel[10] proposed that $e_{t+1} = E + \alpha e_t$, where $\alpha$ is the proportionality constant between errors in the synthetic apparatus built at the previous timepoint and errors in proteins that are newly synthesized (in the next timepoint). The error catastrophe occurs if $\alpha \geq 1$ (ref. 10, but see Discussion), since translation error rate ($e_t$. See $y$ axis of Fig. 1a), and consequently the mortality risk (gray scale in Fig. 1a), increases indefinitely with $t$. Without loss of generality, we assumed a critical level of mortality risk with an aggregate error rate of $D$, so that a theoretical maximum lifespan could be expressed as the time taken for $e$ to increase from $e_0$ (= $E$) to $D$ ("Maximum lifespan" in Fig. 1a, see "Methods"). Obviously, in a population, individuals/samples with a higher or lower baseline error rate $E$ will respectively have shorter or longer maximum lifespan (Fig. 1a, b, red and blue symbols, respectively). Thus, if the maximum lifespan were not affected by factors other than the translation error, we should predict a perfect anticorrelation between maximum lifespan and the basal translation error rate as shown by the lifespan-to-error-rate curve in Fig. 1b. There may, however, be deaths due to other causes before the maximum lifespan that translation error dictates. Therefore, our model predicts a maximum lifespan based on the translation error rate, or, in other words, that lifespan of different samples will fall below the lifespan-to-error-rate curve in Fig. 1b. This can also be explained more intuitively as the "buckets effect" - the maximum practical capacity of a bucket with staves of unequal length is limited by the length of the shortest stave. In the same way, samples below the lifespan-to-error-rate curve in Fig. 1b represent deaths caused by genetic factors other than the Error-Catastrophe, such as telomere attrition, genomic instability, mitochondrial dysfunction, etc.

Now, let us consider the narrow range of variation in translational fidelity due to pleiotropic constraint and/or drift barrier (Fig. 1c), which can be represented by a lower ($L$) and an upper ($U$) limit to $E$ ($x$ axis of Fig. 1c). Accordingly, the lifespan of different individuals in a population can be represented by samples (red dots) randomly distributed within the green trapezoid (with a concave hypotenuse) region (Fig. 1c). One of the most notable features of this presentation is the intuitive demonstration of how the limited variation of translational fidelity obscures the detection of the fidelity-longevity correlation. In particular, the green trapezoid consists of the relatively short-lived samples within a rectangle (light green region in Fig. 1d) and the relatively long-lived samples within a right triangle with a concave hypotenuse (dark green region in Fig. 1d). Obviously, the fidelity-longevity correlation is evident only among the long-lived samples within the triangle, but not among the short-lived samples within the rectangle. From a biological perspective, this is similar to cancer being one of the predominant causes of mortality and morbidity among the elderly, but not among the young[51,52]. Most importantly, our modeling suggests a simple workaround - excluding some short-lived samples from the analyses should enrich samples in the triangle and therefore increase the power of statistical tests for the Error-Catastrophe. In other words, the fidelity-longevity correlation should increase gradually as we remove short-lived individuals from the study (Fig. 1d), until the sample size becomes too small for meaningful statistical analysis.

To further demonstrate the validity of our logic, we conducted an in silico simulation according to the above model with realistic parameters of $\alpha = 1.5, D = 0.06, L = 5 \times 10^{-4}$ and $U = 2 \times 10^{-3}$ (see "Methods"). No significant correlation between lifespan and translational fidelity can be found when we use all samples generated from the trapezoid region defined by these parameters (Fig. 1e, leftmost bar). The correlation, however, increased when some short-lived samples were removed. And as a consequence, we were able to obtain a significant fidelity-longevity correlation after removing 40% short-lived samples (Fig. 1e. Bonferroni adjusted $P < 0.05$ in

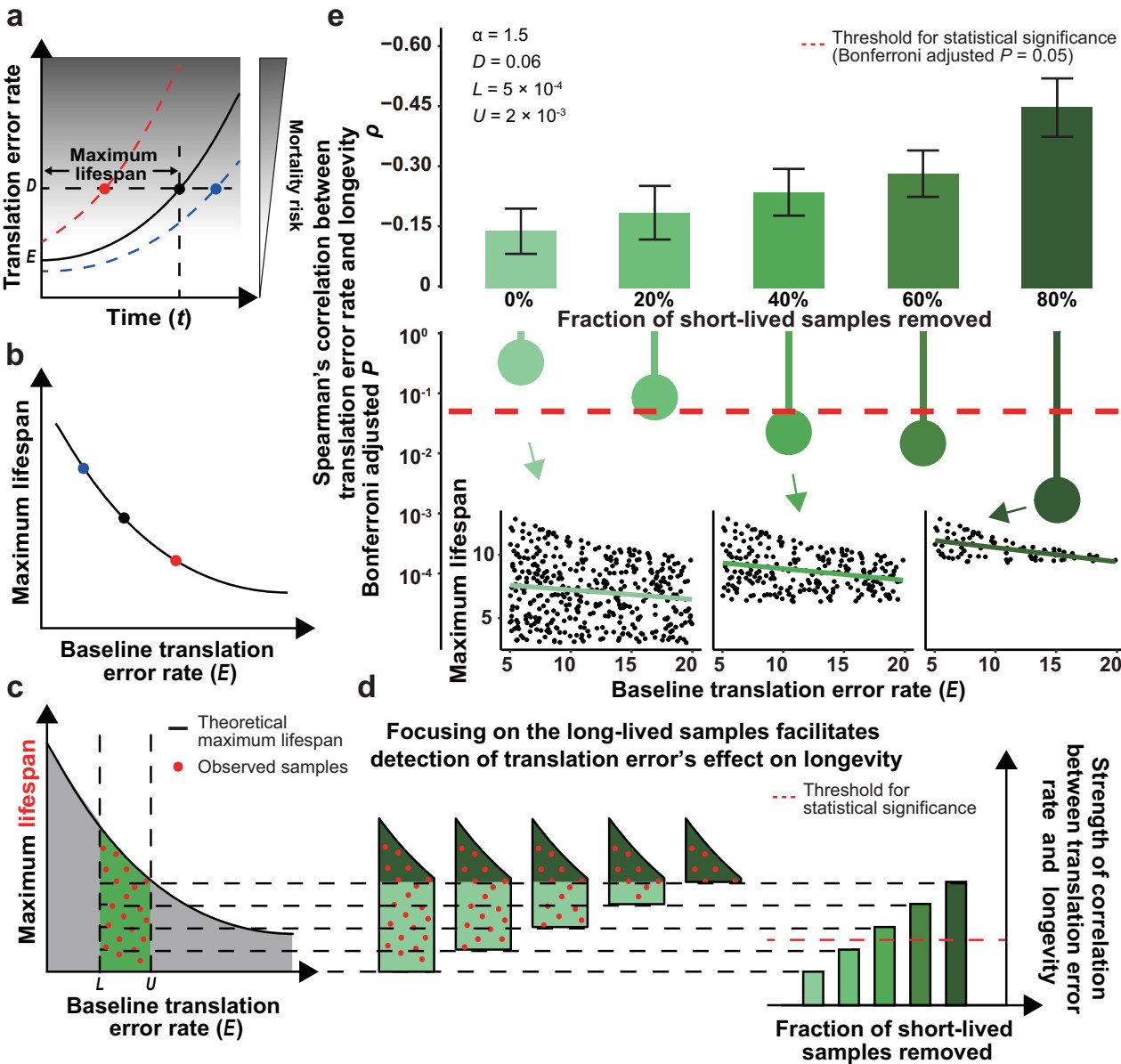

**Fig. 1 | The error-catastrophe concealed by the limited variation of translational fidelity. a** Translation error rate (*y*) increases with time (*x*). *E* is the baseline translation error rate, while *D* is the translation error rate corresponding to a critical level of mortality risk. The time it takes for the translation error rate to increase from *E* to *D* determines the theoretical maximum lifespan. There are samples with higher and lower baseline translation error rates presented in differentially colored symbols. **b** The relationship between baseline translation error rate and maximum lifespan as determined by the model in **a**. **c** The expected lifespan of any genotype can only be lower than the theoretical maximum lifespan. Since the translation error rate is evolutionarily constrained, samples are further confined within the green trapezoid region, and cannot appear in the grey region. **d** The effect of translation error on lifespan is only evident in the dark green triangle but not in the light green rectangle. The removal of short-lived samples

(black dashed lines) should enrich samples in the triangle, and thus enhance the correlation between translation error rate and longevity beyond statistical significance. **e** The correlation between translation error rate and longevity in computationally simulated samples. Spearman's Correlation Coefficient *ρ* and corresponding Bonferroni adjusted *P* values are shown in the upper and lower halves, respectively. There is a stronger correlation when a fraction (*x*) of the short-lived samples is removed from the analysis. Scatter plots with linear regression models in green lines are shown for three specific simulation results as insets (n = 400, 240, 80, from left to right). The mean and standard deviation of Spearman's *ρ* across 1000 simulations are indicated by the bars and error bars, respectively. Key model parameters underlying these results are also listed at the top left corner. See also Supplementary Fig. 1 for simulations conducted with varied parameters.

Spearman's Rank Correlation Test). After varying each individual parameter (α, *D*, *L*, *U*) to a range of three-fold differences around the aforementioned realistic values, we found that such enhanced signal of Error-Catastrophe following the removal of short-lived samples was robust to parameter selection (Supplementary Fig. 1). In conclusion, to test the fidelity-longevity correlation, analyses focusing on long-lived samples are required due to the limited variation of translational fidelity.

**Experimental assessment of chronological lifespan and translational fidelity in a panel of yeast recombinant haploid progeny**

With the above theoretical considerations in mind, we utilized a panel of *S. cerevisiae* recombinant haploid progeny[53] derived from a cross between BY and RM to test the correlation between translational fidelity and longevity. The segregant strains, on the one hand, have diverse phenotypes and, on the other hand, have been genotyped[53] so that QTL analyses can be conducted to identify the genetic factors

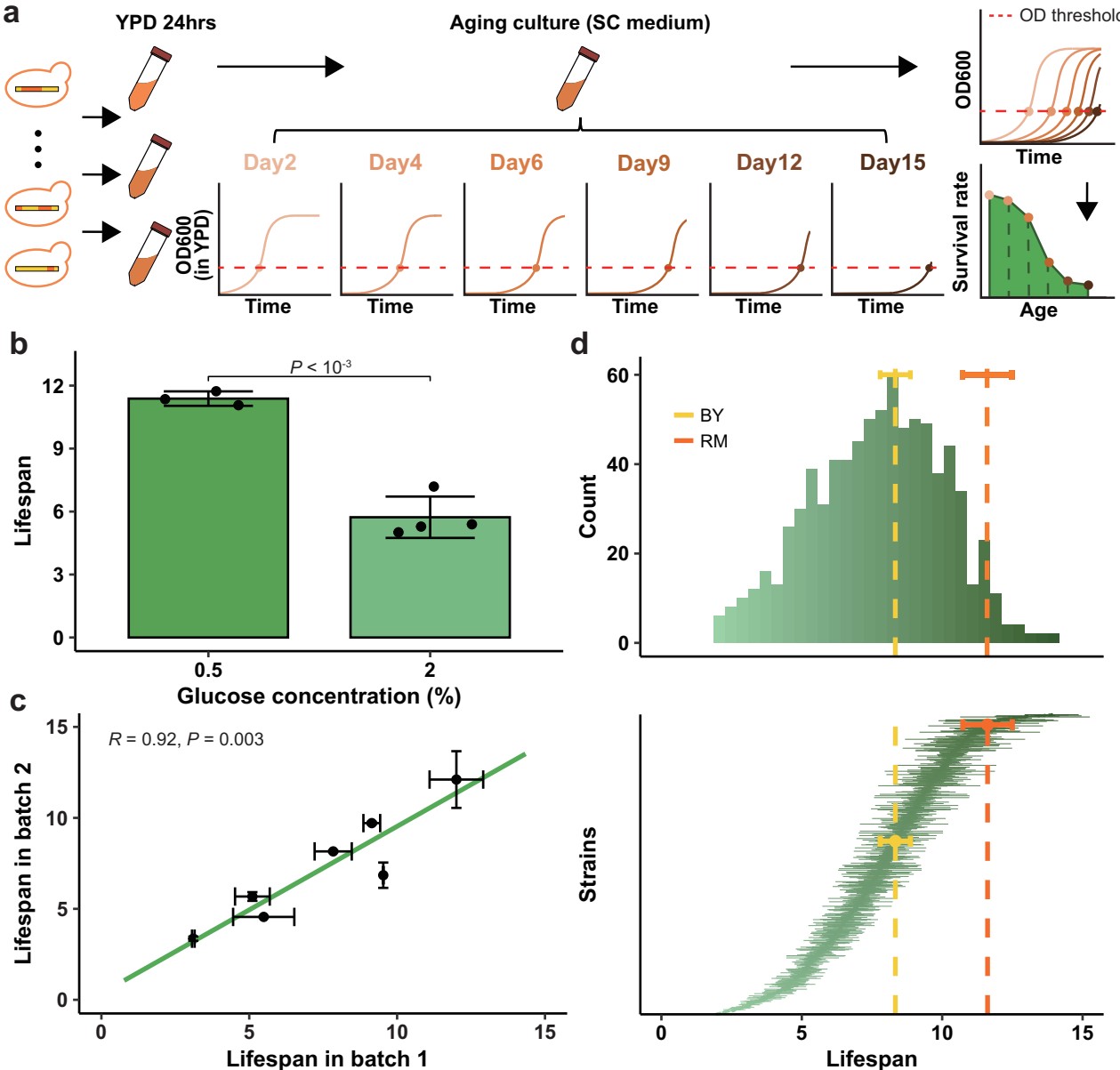

**Fig. 2 | Experimental assessment of chronological lifespan. a** Overview of the experimental pipeline. The strains were inoculated into YPD medium for 24 hours of cultivation, and then transferred to SC medium for a 15-day aging culture. The relative survival rates of cells on days 2, 4, 6, 9, 12 and 15 were estimated by transferring a small (5 μL) subculture to YPD medium and assessing the time at which the growth of the subcultures exceeds a certain OD threshold. Finally, the chronological lifespan was estimated by the area under the age-survival rate curve. **b** Our experimental pipeline successfully recapitulated the known lifespan-lengthening effect of carbon restriction. Chronological lifespans of BY in 0.5% (n = 3) and 2% (n = 4) glucose YPD media were individually shown as dots, with their average value and standard deviation respectively indicated by the bars and error bars (one-tailed Student's *t*-test P = 2.5 × 10$^{-4}$). **c** Estimated lifespans are highly reproducible. The chronological lifespans of segregant strains were assessed in two batches. Seven randomly chosen strains were assayed in both batches, which

appears highly consistent. No significant between-batch difference was detected by the Wilcoxon rank-sum test for each strain. Pearson's Correlation Coefficient *R* and corresponding *P* values from two-tailed Pearson's correlation test are indicated. Each dot represents the average lifespan of three biological replicates, with horizontal and vertical error bars represent standard deviation of three biological replicates in batch 1 and batch 2, respectively. **d** Overview of the 804 chronological lifespans obtained. The upper panel displays a histogram for the distribution of lifespans averaged across three replications per strain. The lower panel shows each strain individually, with a dot representing the average lifespan and a horizontal error bar depicting the standard deviation across replications. The strains were sorted by increasing average lifespan from bottom to top. Dashed line and error bar respectively represent the average and standard deviation of parental strain lifespans, using colors indicated by the legend.

influencing translational fidelity and longevity. Having sequenced ten randomly selected strains and confirmed their genotypes agree with previous reports[53,54] (Supplementary Fig. 2), we sought to measure these strains' chronological lifespans using a previously published high-throughput method[55]. In brief, each strain is cultured on synthetic complete (SC) medium for 15 days of aging culture. The relative survival rates of cells on days 2, 4, 6, 9, 12 and 15 were determined by the

growth curve of a subculture on YPD seeded by 5 μL of the aging culture (Fig. 2a, see "Methods"). A strain with a longer chronological lifespan would have a larger area under its age-survival rate curve.

Several measures were taken to ensure our experimental pipeline's accuracy. First, we confirmed that our 15-day aging culture provides sufficient resolution to detect the known[56] lifespan-lengthening effect caused by carbon restriction from 2% to 0.5% glucose in YPD

culture (Fig. 2b). Second, we found that chronological lifespan measurements across batches were highly replicable across seven randomly selected strains (Fig. 2c). Thirdly, three parameter sets previously used[55–57] in estimating the relative survival rates were tested, and the resulting chronological lifespan was highly consistent (Supplementary Fig. 3, see "Methods"). Thus, we concluded that our experimental pipeline can produce accurate and robust estimates of chronological lifespans.

By applying our experimental pipeline, we obtained reliable estimates of chronological lifespan for 804 strains, each replicated at least thrice (see Supplementary Data 1). The chronological lifespans are normally distributed (Kolmogorov-Smirnov test, $P = 0.7141$) (Fig. 2d upper panel). Based on an Analysis of Variance (ANOVA), the differences between replicates are much smaller than the differences between strains, such that strain identity explains a significant fraction of the variance in lifespans ($F = 33.20$, F-test $P < 10^{-15}$) (Fig. 2d lower panel). An upper bound for the total contribution of genetic differences between strains to lifespan variation (broad-sense heritability, or $H^2$) was estimated by the fraction of total variance in lifespans explained by the repeatability of measurements for each strain, which is 92.00% (S.E. = 0.65%). Similarly, the fraction of lifespan variance explained by the additive effects of all segregating markers (narrow-sense heritability, or $h^2$) was estimated as 33.08% (S.E. = 5.68%), which sets an upper bound for the total amount of additive genetic variance that could be explained with a QTL-based model.

Next, a dual luciferase reporter system was used to quantify the translation error rate of the segregant strains[34,35] (Fig. 3a). This system contains a transgene encoding two luciferases, Firefly and Renilla, as a fusion protein, which allows measurement of concentration-independent Firefly activity based on the ratio between Firefly and Renilla luminescent signals. We used two versions of Firefly, a wildtype with codon AAA encoding Lysine at position 529 and a mutant with codon AGG encoding Arginine at the same position. It has been shown that the mutant will only exhibit Firefly activity if the protein is mistranslated to Lysine at position 529, because no other side chain interacts with the luciferase substrate as does the Lysine side chain[25,34]. Consequently, the Firefly activity of the mutant, relative to that of the wildtype, measures the translation error rate[34,35]. Note that the translation error rate measures both the effect of synthetic error and the sorting/degradation of erroneous proteins. We obtained reliable estimates of translation error rate for 260 segregant strains, each replicated at least thrice. The quality of this dataset was furthered ensured by resequencing the strains carrying the chromosome-bound dual luciferase reporter (Supplementary Fig. 4), as well as largely consistent Firefly to Renilla ratios of luminescent signals for wildtype Firefly-carrying strains (Fig. 3b) and for mutant Firefly-carrying strains (Fig. 3c). The translation error rates are overall normally distributed (Kolmogorov-Smirnov test, $P = 0.5616$. Figure 3d upper panel). According to an ANOVA, the differences between replicates are much smaller than the differences between strains, suggesting that strain identity explains a significant portion of the variance in translation error rates ($F = 7.307$, F-test $P < 10^{-15}$) (Fig. 3d lower panel). The broad-sense and narrow-sense heritability of translation error rate was estimated as 72.92% (S.E. = 3.18%) and 12.17% (S.E. = 12.69%), respectively.

### Translational fidelity is correlated with longevity among long-lived strains

We then tested the fidelity-longevity correlation among the 235 segregant strains with both translation error rate and chronological lifespan data (Supplementary Data 2). Upon analysis of all 235 strains, no significant correlation could be found (Spearman's $\rho = -0.032 \pm 0.069$, $P = 0.629$. The left-most bar in Fig. 4a and b). However, when short-lived strains were removed, the anticorrelation between translation error rate and lifespan gradually increased (Fig. 4a and b). Indeed, after removing 50%, 70%, and 90% short-lived strains, Spearman's $\rho$ for the

fidelity-longevity correlation became $-0.198 \pm 0.095$, $-0.284 \pm 0.137$, and $-0.551 \pm 0.139$, respectively, all with nominal $P < 0.05$. After Bonferroni adjustment for multiple testing, the correlation after removing 90% short-lived strains remained statistically significant ($P < 0.036$). This observation is exactly expected by our theoretical model that takes into account the limited variation of translational fidelity (Fig. 1). As a result, the association between translational fidelity and longevity is directly supported by the strong anticorrelation observed between translation error rate and lifespan among long-lived strains.

### The gene *VPS70* underlies the correlated variation of longevity and translational fidelity

In order to further investigate the genetic basis of the fidelity-longevity correlation, we performed QTL mapping for chronological lifespan and translation error rate for the top 50% ($n = 118$) long-lived strains, which is a compromise between the strength of fidelity-longevity correlation (nominal $P < 0.05$) and detection power (sample size) of QTL mappings for both traits. As for chronological lifespan, we found significant linkage at one locus on chromosome X (Fig. 5a, b, green symbols). Similar analysis of translation error rates found significant linkages at two loci on chromosomes X and III (Fig. 5a, b, blue symbols). Intriguingly, the most significant SNPs in both analyses are only 3086 base pairs apart on chromosome X, and their confidence intervals of the QTL overlap (Fig. 5b). Therefore, this overlapped region on chromosome X from 641,753 to 669,427 represents a promising candidate for the genetic factor that drives the correlated variance of longevity and translational fidelity. We examined the one SNP with the largest combined LOD score (chrX:655,475) within this overlapped region, and found that RM alleles (Thymine) confer longer lifespans (Fig. 5c) and lower translation error rates (Fig. 5d) than BY alleles (Cytosine). Note that the RM parental strain itself showed a higher translation error rate than the BY parental strain (Fig. 3d), which might be explained by other genetic factors independently affecting translational fidelity and/or longevity (Supplementary Fig. 5).

We tested individual genes containing nonsynonymous SNPs between RM and BY in this overlapping region, which include *ILM1*, *JHD2*, *STE24*, *VPS70* and *RSF2-YJR128W* (two reverse complementarily overlapping ORFs). A two-step method[58,59] was used to replace each of these genes in the BY parental strain with the same genes in the RM parental strain. It was only when *VPS70* was replaced by its RM allele (the VPS70-RM allele) that a significant lifespan extension of ~8.9% (median value 7.05–7.66) was observed relative to the BY parental strain (Fig. 5e, the BY::VPS70-RM strain). More importantly, the BY::VPS70-RM strain also showed ~8.0% (median value 0.000849–0.000781) reduced translation error rate (Fig. 5f). Interestingly, it has been shown that the deletion of *VPS70*, which encodes the Vacuolar Protein Sorting-associated protein 70, reduces competitive fitness[60] and increases chronological lifespan[61].

We then aimed for a preliminary exploration on the mechanism by which VPS70 reduces translation errors and extends lifespan. Despite the current lack of a specific molecular mechanism to explain the function of VPS70, it is widely known that vacuoles play an important role in lifespan determination through the enzymatic breakdown and recycling of cellular waste[62]. Our findings led us to hypothesize that VPS70 regulates the vacuole-dependent processing of mistranslated proteins, thereby influencing the amount of mistranslated proteins in a cell. In order to test this hypothesis, we treated both the BY and BY::VPS70-RM strains with Concanamycin A (ConA), which inhibits vacuolar ATPase, neutralizes the pH of the vacuole, and blocks trafficking into the vacuole and inhibits degradation in the vacuole[63–65]. Treatment with ConA abolished both the translation error reduction and the lifespan-lengthening effect conferred by the RM allele of *VPS70* (Fig. 5g). It is clear from this observation that VPS70 regulates translation error rate and lifespan in a vacuolar-dependent manner. Similarly, treatment with rapamycin, which lengthens lifespan and enhances

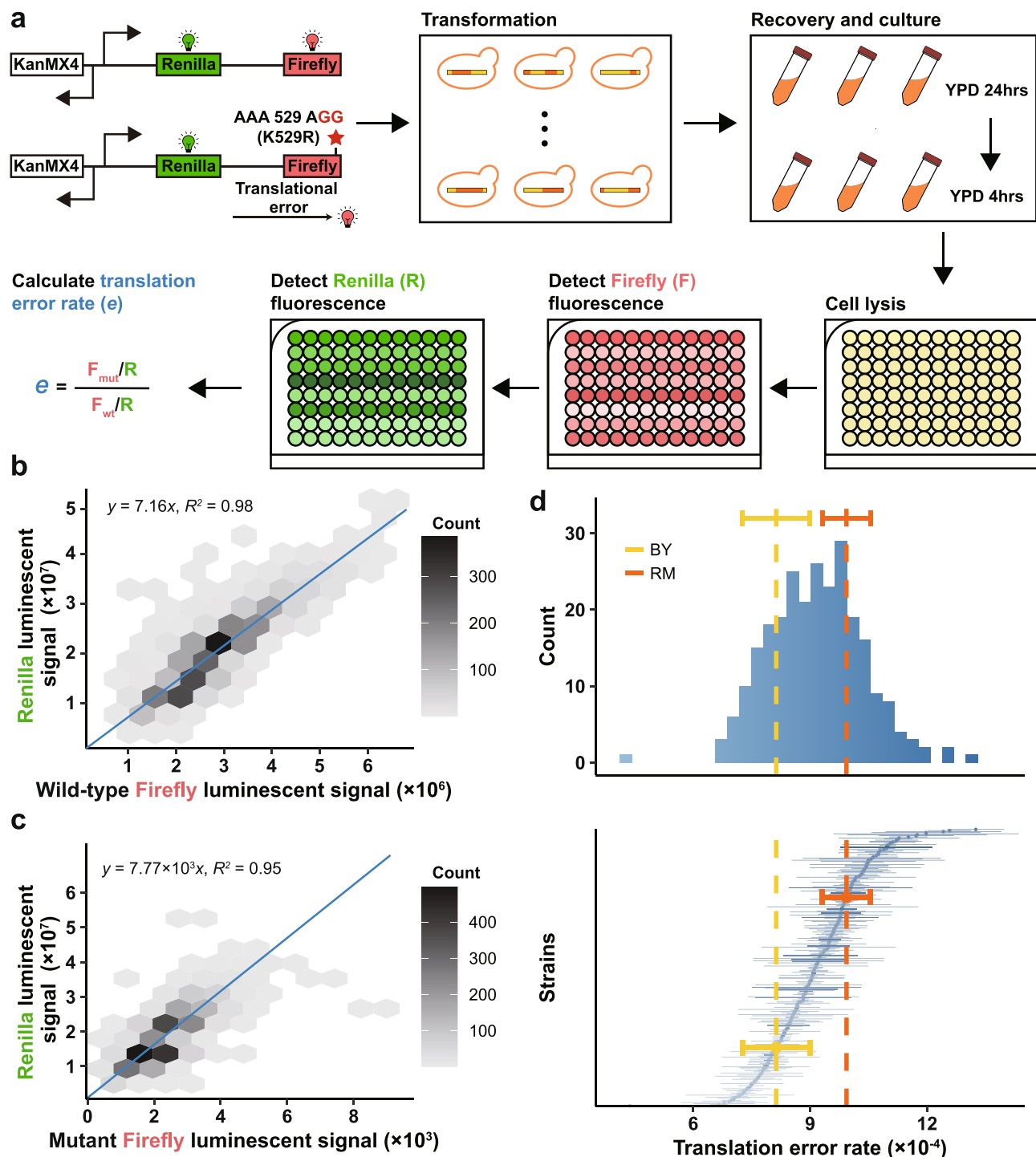

**Fig. 3 | Experimental measurement of translation error rate. a** Overview of the experimental pipeline. A dual luciferase system consisting of Renilla and either wildtype or mutant Firefly was transformed to the segregant strains. Each strain was cultured and assayed for Firefly and Renilla activities, which were used to estimate translation error rate (See "Methods"). **b** The Firefly to Renilla ratios of luminescent signals are largely consistent across strains carrying the wildtype Firefly. The distribution of data points representing the Firefly (x) and Renilla (y) luminescent signals of individual replicates was shown as a hexagonal binning plot, with the colors representing the number of points falling in each hexagon by the color scale to the right. A diagonal line shows the linear model fitted to the raw data, whose parameter and Coefficient of Determination ($R^2$) are indicated in the top left corner. **c** Similar to **b** except that the strains carrying the mutant Firefly are analyzed. **d** Similar to Fig. 2d except translation error rates are analyzed.

translational fidelity through affecting the mTOR pathway[30,66], eliminates the effect of VPS70-RM, suggesting that function of VPS70 is dependent on the mTOR pathway as well (Supplementary Fig. 6). Combined, these results further confirm *VPS70*'s significance as a genetic link between translation error rate and longevity.

## Discussion

In this study, we provide direct genetic evidence that translational fidelity contributes to intra-specific variation of longevity. Specifically, we showed that if translation error rate is tightly constrained within a limited range, the correlation between translational fidelity and

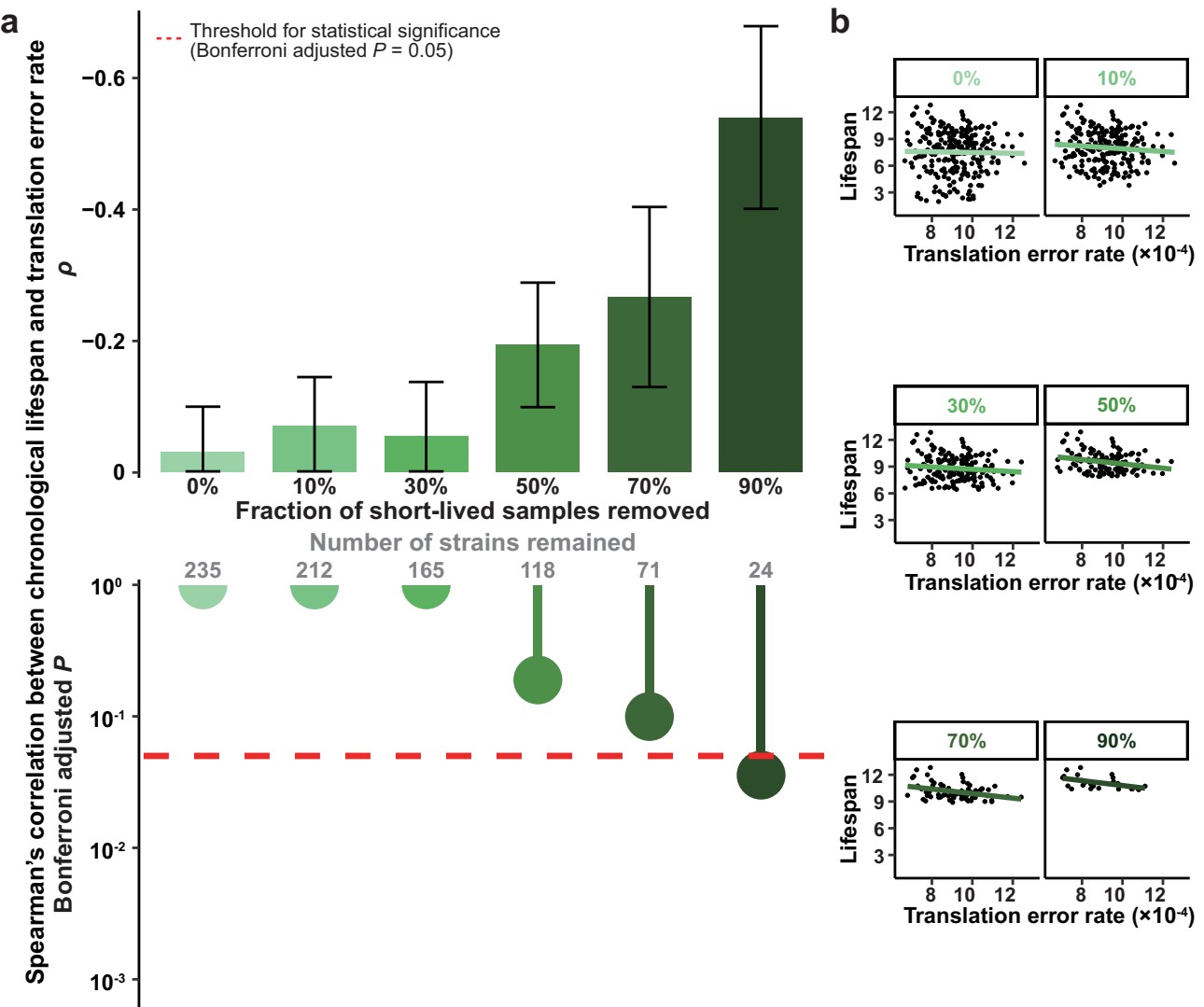

**Fig. 4 | Correlation between chronological lifespan and translation error rate.**
**a** The Spearman's Correlation Coefficient $\rho$ between chronological lifespan and translation error rate was calculated after a fraction (x axis, the number of strains remained is also shown) of short-lived strains was removed from the 235 strains. The bars in the upper panel indicate Spearman's $\rho$, with the error bars representing the standard error of the mean assessed by 100 bootstraps, whereas the lollipops in the lower panel indicate the statistical significance from two-tailed Spearman's correlation test, with the threshold of Bonferroni adjusted $P = 0.05$ marked by the dashed line. **b** Scatter plots with linear regression models in green lines are shown for each fraction in **a**.

lifespan will be obscured, and only detectable by focusing on long-lived samples. By experimental measurement of both the lifespan and translational fidelity in a panel of 235 strains from the *Saccharomyces cerevisiae* BY × RM recombinant haploid progeny, it was confirmed that greater translational fidelity was indeed associated with an increased lifespan in the long-lived strains. A QTL analysis of this subpopulation identified a locus on chromosome X that was significantly linked with both translational fidelity and lifespan, which encompasses the gene *VPS70* (Vacuolar protein sorting-associated protein 70). We found that the RM allele of *VPS70* reduced translation error rate and prolonged the lifespan of the BY strain through a mechanism dependent on the normal function of vacuoles. Collectively, these results strongly suggest that translation error contributes to intra-specific differences in longevity, particularly among long-lived individuals.

Our study had potential caveats that warrant discussion. First, our observations might not necessarily be explained solely by Orgel's hypothesis, as other confounding factors may be increasing translational fidelity and extending lifespan at the same time. Growth rate, for example, may be one such confounding factor, as strains with slower growth tend to live longer[67,68] due to mechanisms such as superior resource management, which might also enhance translational fidelity. However, the Error-Catastrophe Theory of Aging still provides the most coherent explanation of the evidence that is currently available. This is because, on the one hand, the confounding by growth rate as outlined above predict a fidelity-longevity correlation regardless the strain's lifespan, meanwhile as reasoned in this study, the Error-Catastrophe Theory of Aging predicts that fidelity-longevity correlations are more apparent among the long-lived strains, which is observed in our empirical dataset. On the other hand, it is not easy to distinguish their causal relationship given the complicated interplay among translational fidelity, longevity, and growth rate. It is possible to argue, for example, that both enhanced longevity and slowed growth are the result of increased translational fidelity. More importantly, such a causal relationship is indeed directly supported by the changes in lifespan following manipulation of translational fidelity[30–32]. Second, a theoretical argument against Orgel's original model is that α, which serves as a proportionality constant between errors in the synthetic apparatus built at the previous timepoint and errors in newly synthesized proteins, is not necessarily greater than 1. Consequently, a

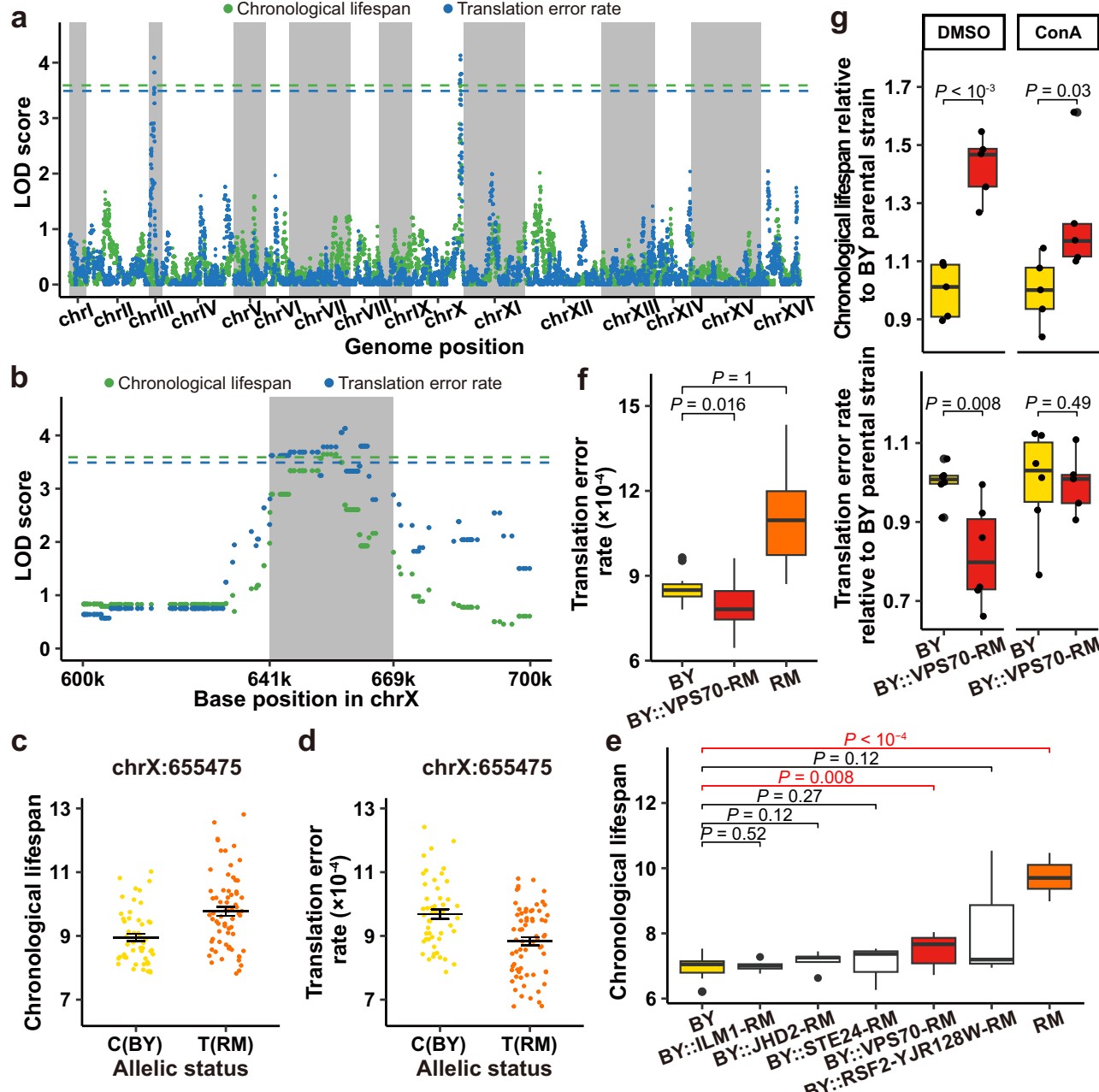

**Fig. 5 | QTL mapping identified *VPS70* as the gene underlying the correlation between longevity and translational fidelity. a** Manhattan plot of whole genome QTL mapping for chronological lifespan and translation error rate, using colors as indicated by the legend on top. The threshold for statistical significance of LOD score is represented by the dashed line of the corresponding color. The alternating gray and white regions indicate different chromosomes. **b** Same as **a**, but only chromosome X is shown, and the gray area represents the overlapped region between the confidence intervals of the significant QTLs. **c, d** Association between the allelic status (*x* axis) and the chronological lifespan (**c**) or translation error rate (**d**) is shown for the one SNP with the highest combined LOD score in the overlapped QTL region (*n* = 118). Error bars represent standard errors of the mean. **e** All genes with nonsynonymous SNPs within the overlapped QTL region were individually tested for their effect on longevity by replacing the gene in the BY parental strain with that in the RM parental strain. The chronological lifespan of BY, RM and the gene-replaced strains is displayed as standard box plots (*n* = 12,

5, 5, 3, 13, 3, 6 from left to right). Box upper and lower bounds represent 75th and 25th percentiles, respectively. The line inside the box indicates the median. Whiskers correspond to maximum and minimum values within 1.5 × interquartile range from the quartiles; points outside of this range are outliers. *P* values from one-tailed Wilcoxon rank-sum tests are shown on top, with significant values highlighted red. **f** Same as (**e**) except that only *VPS70* is tested for its impact on translation error rate (*n* = 14, 15, 18 from left to right). **g** ConA treatment, which inhibits degradation in the vacuole, eliminates the translation error rate-reducing (lower panel; *n* = 6, 5 from left to right) and lifespan-lengthening effects (upper panel; *n* = 5, 5) conferred by the RM allele of *VPS70* over its BY parental strain (*y* axis, scaled so that the average of BY equals 1). A parallel treatment with DMSO is used as a control (*n* = 6, 6 for translation error rate and *n* = 5, 5 for lifespan; *P* = 0.008 and 1 × 10⁻⁴). The results are presented as standard box plots with the same elements as in **e**, except that all original points are shown. *P* values from one-tailed *t*-tests are shown.

catastrophe of ever increasing error rate will not occur[10], as the error rate will only increase logarithmically to a limit of $\frac{E}{1-\alpha}$. The link between excessive translation error and death may still be valid, however, as long as this maximal translation error rate is still too high for proper cellular function without senescence, or the value of α increases with age. Third, although we follow previous practices to approximate aging by longevity in a model organism, aging and longevity do not necessarily correspond[69]. Lastly, the specific mechanism by which *VPS70* function is still unclear, so future research will be required.

With genome-wide QTL mapping in the long-lived strains, we found the loci with strong associations with both translational fidelity and lifespan on chromosome X, suggesting that the genetic factors linking the two traits are primarily located on this chromosome. Our subsequent experiments tested each candidate gene individually and narrowed down the key factor to *VPS70*, a gene known to be involved in vacuolar protein sorting. From a mechanistic perspective, it has previously been demonstrated that vacuoles are functionally linked to biological processes that recycle damaged/erroneous proteins[62], such as autophagy, a known anti-aging mechanism[70,71]. Changes in vacuolar pH also affect lifespan in yeast[62,72]. Thus, our identification of *VPS70* simultaneously impacts translational fidelity and longevity suggests that mistranslated proteins may be degraded more efficiently in long-lived strains via a vacuole-dependent process, which is indeed supported by our preliminary experiments (Fig. 5g) with ConA, an inhibitor of protein degradation in the vacuole[63–65]. Orgel speculated in his original paper about how to avoid error catastrophe in a single cell. "What is needed is a selection based on the accuracy of protein synthesis, that is, a selection which rejects enzymes which lead to too many errors in protein synthesis. This could be achieved within a single cell only by a partial or complete segregation of the products of one piece of 'protein synthetic apparatus' "[9]. It is possible that the interaction between VPS70 and ribosomes[73] is selective for the mistranslating ribosomes, so that the mistranslated peptides and/or the mistranslation-prone ribosomes would be directed to the vacuoles for rapid degradation. Clearly, more functional studies are required.

One key conceptual advance of our study is the discovery that the fidelity-longevity correlation is obscured by the limited variation of translational fidelity. Besides allowing us to design more elaborate tests for the role of translational fidelity in intra-specific lifespan variation (e.g. focusing on the long-lived samples), this finding has profound theoretical ramifications. For example, if natural selection has largely been reducing translation error rate for the sake of fitness, as excessive translation errors are harmful[35,37–39], the extended lifespan as a result of the fidelity-longevity correlation can then be considered to be a by-product of the direct natural selection for increased fidelity. This notion stands in contrast with the antagonistic pleiotropy theory of aging[74], in which selection favors early advantage for reproductive success but comes with a cost at the later stage of life (increased death risk and therefore shortened lifespan).

Although the present study focuses on translational fidelity and lifespan, pleiotropy may similarly obscure the correlation between any trait pair. Using a heuristic approach, we were able to enrich samples with better representations of the correlation between the two traits (Fig. 1d). Alternative methods may, however, be available for identifying correlated traits as well as the underlying QTLs. Taking our dataset as an example, when all yeast strains with available data were analyzed, the peak on chrX remained the highest in the lifespan QTL analysis (Supplementary Fig. 7a, 804 strains for CLS and 260 strains for translation error rate), but was not associated with translation error rate, and there was no correlation between longevity and translational fidelity (Supplementary Fig. 7b). Rather than removing 50% of the shortest-lived samples, the correlation between longevity and translational fidelity could also be demonstrated by controlling the second (Supplementary Fig. 7c) and third (Supplementary Fig. 7d) most significant lifespan QTLs. Specifically, when we focused our analyses on

strains with Thymine on chrXII:660,371 (the RM allele associated with longer lifespan) and Guanine on chrXIV:461,485 (the RM allele associated with longer lifespan), longevity and translational fidelity exhibit significant correlations (Supplementary Fig. 7e) as well as overlapping QTL peaks (Supplementary Fig. 7f), although both peaks fail to achieve statistical significance. On the contrary, the fidelity-longevity correlation is absent among strains with the other alleles on either or both strains (Supplementary Fig. 7g-i). These two loci were examined more closely and it was discovered that the chrXIV:461,485 position is located within the ORF YNL088W, which encodes Topoisomerase II, a protein that relieves torsional strain in DNA by cleaving and resealing phosphodiester backbones of both positively and negatively super-coiled DNA. The chrXII:660,371 position does not correspond to any ORF, but a 10kbp region surrounding it contains several known genes, including *YLR257W* (encoding a protein of unknown function), *YLR258W* (a glycogen synthase), and *YLR259C* (tetradecameric mito-chondrial chaperonin). The development of more general statistical methods for the systematic detection of similarly obscured phenotype-phenotype correlations or genotype-phenotype associations (or cryptic QTL) might be worthwhile in the future, such as the most recent effort on the level of a chromosome by epic-QTL[75].

## Methods

### Simulations based on the Error Catastrophe Theory of Aging

On the basis of the mathematical model proposed by Leslie Orgel for propagation of translation error[10,50], we denote the aggregate translation error rate at time $t$ as $e_t$, the baseline translation error rate as $E$, and therefore have $e_{t+1} = E + \alpha e_t$, where α is the proportionality constant between errors in the synthetic apparatus built at the previous timepoint and errors in proteins that is newly synthesized. It can be shown that when $\alpha \geq 1$, $e_t = E(1 - \alpha^t)/(1 - \alpha)$ increases indefinitely with $t$, resulting in an error catastrophe. If we further assume an aggregate error rate of $D$ corresponding to a critical level of mortality risk, the life expectancy of an individual can be expressed as the time it takes for $e$ to increase from $e_0$ (= $E$) to $D$, which can be derived as follows:

$$Lifespan = \frac{\log\left((\alpha - 1)\,D + E\right) - \log\left(E\right)}{\log \alpha} \tag{1}$$

The translation error rate is evolutionarily constrained between a lower and an upper limit, which are denoted $L$ and $U$, respectively. For the purpose of simulating the fidelity-longevity relationship under the Error-Catastrophe Theory, a core set of empirically derived values for the four key parameters was used first, i.e. $\alpha = 1.5$, $D = 0.06$, $U = 1.5 \times 10^{-3}$ and $L = 5 \times 10^{-4}$. Specifically, α is selected as a conservative value fulfilling Orgel's original proposition ($\alpha > 1$)[9,10]; $U$ and $L$ are derived from previous luciferase-based[34] and mass-spectrometry-based[36] measurements of translation error rates in yeast; $D$ is chosen to be slightly higher than the highest translation error rate observed in microbes treated with rifampicin[76]. To explore potential inaccuracies of the above parameters and to test the robustness of the observed pattern, each parameter was individually varied by three-fold. More specifically, we tested $U = 1 \times 10^{-3}$ to $3 \times 10^{-3}$, $\alpha = 1.25$ to $1.75$ (i.e., 0.25 to 0.75 beyond the baseline of $\alpha > 1$), $D = 0.03$ to $0.09$ and $L = 2.5 \times 10^{-4}$ to $7.5 \times 10^{-4}$ (Fig. 1e and Supplementary Fig. 1). For each single simulation conducted for a given parameter set, 400 random samples were generated following a bivariate (lifespan and baseline translation error rate) uniform distribution within the possible value range (the cyan trapezoid with a concave hypotenuse in Fig. 1c). For each parameter set, the simulation was performed 1000 times to determine the mean and standard deviation of Spearman's ρ between translation error rate and lifespan.

## Yeast recombinant haploid progeny

All 1056 strains of the yeast BY × RM recombinant haploid progeny (segregants), along with their genotype data, were kindly provided by the authors of previous reports[53,54]. These strains and their correspondence with the genotype data were tested for quality at three different levels. First, we cultured all the strains in yeast peptone dextrose (YPD) medium with 100 µg/mL nourseothricin (NTC, Solarbio). One entire plate of 96 strains, and 44 strains from various other plates, totaling 140 strains, showed resistance inconsistent with previous reports[53,54], and were thus discarded (Supplementary Data 1). Second, we scanned the genotype data to identify for each strain a 1-kbp region that is unique to the strain. We then Sanger-sequenced these unique regions for 50 randomly chosen strains and, as a result, verified their identity. Finally, 10 strains were selected at random and subjected to whole genome DNA sequencing by Illumina platform (raw data has been deposited under the accession number PRJNA750521 in NCBI BioProject). We examined all reads aligned to each segregating site to determine the fraction of reads that are consistent with the genotype data (Supplementary Fig. 2). Among all 28,220 segregating SNPs in these 10 strains, 95.8% of the sites have >90% reads consistent with the genotype data. As a control, we randomly chose ten strains from the segregants as fake references, and similarly calculated the fraction of reads consistent with the genotype data, which was then averaged across all segregating sites in all ten strains. Such average fractions were repeatedly calculated 1000 times to gauge the null distribution (Supplementary Fig. 2). As a precaution against genotype changes caused by the manipulation of the luciferase genes, 260 strains carrying the dual luciferase system were also genotyped (Supplementary Fig. 4). Since these genotyping efforts suggest that the previously listed genotypes for these 260 strains are accurate, we have opted to stick with the original genotype lists throughout our entire study.

## Measurement of Chronological Lifespan

To determine the chronological lifespan (CLS) of the yeast strains, we used a previously proposed high-throughput method[55]. In particular, all strains were inoculated onto YPD solid medium, then transferred to YPD liquid medium for overnight cultivation after the formation of a single clone. Three biological replicates of each resuscitated strain were then transferred to synthetic complete (SC) medium for 15 days of aging culture. On days 2, 4, 6, 9, 12, and 15 of the aging culture, 5 µL of each experimental replicate was transferred to a 96-well plate containing 145 µL of YPD medium to measure the fraction of surviving cells. For this purpose, we used a microplate spectrophotometer (Epoch2, BioTek) to measure the OD600 every 10 minutes for a period of 12 hours at 30 °C, without shaking the plates. After subtracting the background OD600 measured from the three cell-free, YPD-only wells on the same 96-well plate, we compared the three technical replicates under each biological replicate to remove outlier or noisy measurements. Briefly, if the coefficient of variation (CV) of OD600 among the three technical replicates at $t = 50$ minutes exceeded 0.2, the one technical replicate that differed most from the other two was removed. If the two remaining technical replicates still exhibited CV greater than 0.2, all technical replicates (and therefore this biological replicate) would be discarded. A value for the biological replicate was calculated by averaging the remaining technical replicates. Using the OD600 of each biological replicate obtained, we calculated the doubling time (DT) using:

$$\delta_n = \frac{\ln 2}{\left(\frac{\ln X_2 - \ln X_1}{t_2 - t_1}\right)} \tag{2}$$

Here, $\delta_n$ represents the DT of a biological replicate on day $n$ of the aging culture, while $X_1$ and $X_2$ are the background-subtracted OD600 at timepoints $t_1$ and $t_2$. Note that only $X_1$ and $X_2$ during exponential growth (OD600 at 0.2 to 0.5) were used. As there are slight variations between DTs estimated using different combination of $t_1$ and $t_2$, we used the minimal DT in our main figures. However, mean DT or median DT give quantitatively similar results (Supplementary Fig. 3).

Subsequent comparisons between wells were started by controlling for potential differences in initial cell density or optical density among wells. To achieve this, for each well, we subtracted the OD600 measured at timepoint $t = 50$ minutes from all OD600 values. These normalized OD600 values were examined for the first timepoint when it reaches 0.3, which is denoted by $t_n$ for day $n$ of the aging culture. In the event that a culture does not reach OD600 of 0.3 within 12 hours, a predicted $t_n$ will be calculated based on the DT ($\delta$) and OD600 ($X$) at timepoint $t = 12$ hours by $t_n = 12 + (\ln(0.3) - \ln(X))\delta/\ln(2)$. The fraction of surviving cells in each aging culture on day $n$ relative to day 2 was then calculated by:

$$S_n = \frac{1}{2^{\left(\frac{t_n - t_2}{\delta_2}\right)}} \tag{3}$$

Then, the CLS of a biological replicate is calculated as:

$$\mathrm{CLS} = \sum_{i=2}^{N} \left(\frac{S_{n_i} + S_{n_{i-1}}}{2}\right)(n_i - n_{i-1}) \tag{4}$$

Here $n_i$ is the $i$th sampled day of the aging culture (i.e. days 2, 4, 6, 9, 12, or 15), and $S_{n_i}$ is the fraction of surviving cells on that day. Here the CLS is also known as the "survival integral" (SI), and is essentially the area under the age-survival curve (bottom right-most panel of Fig. 2a). Finally, outliers or noisy measurements among the three biological replicates were filtered based on the threshold of CV < 0.2, giving rise to CLS data for a total of 804 segregants (Supplementary Data 1).

## Translation error rate

By using standard yeast transformation protocol[77], we integrated the dual luciferase reporter system, coupled with kanMX as the marker, into the HO locus of the yeast strains. The integrations were confirmed by PCR, and three clones were selected as biological replicates. After resuscitating the transformed strains in YPD for 24 hours, cells were collected by centrifugation at $1699 \times g$ (4000 rpm), washed once, and then resuspended in 250 µL ice-cold phosphate buffered saline (PBS). Cells were then lysed by vortexing with glass beads for 10 min, followed by centrifugation at $1699 \times g$ (4000 rpm) at 4 °C. Using 10 µL of the supernatant, the relative luminescence units (RLUs) of Firefly and Renilla luciferase were measured for 10 seconds with a microplate luminometer (GloMax, Promega) according to the manufacturer's instructions. All Renilla and Firefly RLUs were within the equipment's linear dynamic range. For each biological replicate, three technical replicates were assayed.

The RLU data were filtered according to a stringent set of criteria in order to ensure accurate measurement of the translation error rate. We used only mutant Firefly RLUs between $5 \times 10^2$ and $10^4$, wild-type Firefly RLUs greater than $10^6$, and mutant and wild-type Renilla RLUs between $5 \times 10^6$ and $10^8$. We then calculated the protein abundance-normalized Firefly activity by dividing Firefly RLU by the Renilla RLU[35]. Outliers or noisy measurements among the three technical replicates were filtered based on the threshold of CV < 0.2. A value for each biological replicate was calculated by averaging the remaining technical replicates. Then the three biological replicates were again filtered by a threshold of CV < 0.2 to remove outliers or noisy measurements before being averaged for strain-specific mutant and wild-type Firefly activities. The translation error rate of each strain was calculated as the ratio between its mutant Firefly activity and its wild-type Firefly activity[35]. The biological replicates of mutant and wild-type Firefly were

also individually used to calculate biologically replicated translation error rates, which were used to estimate the standard deviation. Finally, strains with translation error rate greater than $10^{-2}$ were removed, leaving 260 strains with estimated translation error rates (Supplementary Data 1). The many strains removed at this stage reflects three technical difficulties in measuring translation error rates. First, translation errors are extremely rare. The detection of translation error therefore requires high sensitivity, which usually comes at the expense of specificity. Second, the final translation error rate is calculated as a ratio of ratios, which magnifies measurement error (or noise). Third, we noticed that some strains tend to exhibit large CVs among technical replicates, such that the CV among technical replicates is positively correlated between biological replicates of the same strain (two-tailed Pearson's correlation test, Pearson's $R > 0.26$, $P < 1.2 \times 10^{-5}$). This may be caused by an increased level of expression noise of tRNA or genes that interfere with the luciferase activity, or by an increased variability in cell lysis efficiency. Due to these considerations, we implemented a strict requirement for the consistency of biological and technical replicates to ensure the validity of our conclusion, despite the omission of many samples.

### Heritability

Broad-sense heritability ($H^2$) and narrow-sense heritability ($h^2$) were calculated by previous methods[53]. Briefly, $H^2$ was calculated by trait values of strains with at least two biological replicates that passed our quality filter above (i.e., CV < 0.2). Two replicates were randomly chosen to unified sample size across strains, and then $H^2$ was estimated as $\sigma_G^2/(\sigma_G^2 + \sigma_E^2)$, where $\sigma_G^2$ was the genetic variance and $\sigma_E^2$ was the error variance, as estimated by the 'lmer' function in lme4 package of R[78]. Narrow-sense heritability was estimated as $\sigma_A^2/(\sigma_A^2 + \sigma_{EV}^2)$, where $\sigma_A^2$ was the additive genetic variance explained by the single nucleotide polymorphisms and $\sigma_{EV}^2$ was the error variance, as estimated by the rrBLUP package in R[79]. Standard errors of $H^2$ and $h^2$ were calculated by delete-one jackknife.

### QTL

We calculated logarithm likelihood ratio (LOD scores) for each genotypic marker and trait as $-n\left(\ln\left(1 - r^2\right)/2\ln(10)\right)$, where $n$ is the number of genotypes and phenotypes, $r$ is the Pearson correlation coefficient between the genotypes and the trait[53]. To estimate significance empirically, assignment of phenotype to genotype was randomly permutated 1000 times[80]. The threshold of significance for each genetic marker and trait is the average LOD score plus three times the standard deviation. The 95% Bayesian credible interval for each QTL was calculated by the 'bayesint' function of qtl package in R[81].

### Yeast strains for functional verification of individual genes

The strains were constructed by a two-step allele replacement method[58,59]. Specifically, we used BSD resistance gene to knock out the corresponding target gene of BY4716 parental strains without dual luciferase, with wild-type or mutant dual luciferase, respectively. According to standard homologous recombination method[82], the above BSD gene was then replaced with a RM-11 genotype knock-in fragment constructed by fusion PCR, and Sanger sequencing was performed to confirm successful knock-in. Finally, we measured the CLS and translation error rate of the reconstructed strains. Sequences of primers used are listed in Supplementary Data 3.

### Concanamycin A treatments

Yeast cells were cultured overnight in YPD medium to OD600 ~ 1.0 and then diluted by fivefold with fresh YPD medium and cultured in the shaker until they reached an OD600 ~ 0.4. The cultures were subsequently supplemented with either the working concentration of 1 μg/ml freshly DMSO-diluted Concanamycin A (ConA) or DMSO alone (control), both at a final concentration of 0.1% DMSO. Then the cultures were allowed to grow for four additional hours, after which the measurement of CLS and translation error rate was conducted as described above.

### Reporting summary

Further information on research design is available in the Nature Portfolio Reporting Summary linked to this article.

## Data availability

The data generated in this study have been deposited to NCBI Bio-Projects under accession number PRJNA750521, and are provided in the Supplementary Data 2.

## Code availability

The Custom R/Python/Perl code used to perform the analyses and generate results in this study is publicly available and has been deposited in GitHub at (https://github.com/BryanZ-27/SI_TE.). A snapshot of the code is provided on Zenodo[83].

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

## Acknowledgements

This work was supported by the National Natural Science Foundation of China (32170635 and 32222018 to X. C.; 32122022 and 32361133555 to J.-R. Y.), the National Key R&D Program of China (grant number 2021YFA1302500 to J.-R. Y.), the Science and Technology Planning Project of Guangdong Province, China (grant number 2024B1212070013 to J.-R. Y.), and the Guangdong Basic and Applied Basic Research Foundation (2022A1515110749 to G. Y.). We thank Jianzhi Zhang, Mengyi Sun, Yi Zhang, Shengbao Suo, Xia Shen, Shuhua Xu, Xionglei He, Vera Gorbunova and Andrei Seluanov for their valuable comments on the manuscript.

## Author contributions

J.-R. Y., X.C. conceived the idea, and designed and supervised the study. W.Z., G.Y., W.S., S.D., Z.Z., Y.S. and W.W. conducted experiments and acquired data. B.Z., G.Y., W.S., X.Z., J.C., Y.S., E.L., X.C. and J.-R. Y. analyzed the data. B.Z. and J.-R.Y. wrote the paper with inputs from all the authors.

## Competing interests

The authors declare no competing interests.
