## [Transparent Peer Review file · Nature Communications]

Translational Fidelity and Longevity are genetically linked

Corresponding Author: Professor Jian-Rong (Philip) Yang

Version 0:

Reviewer comments:

Reviewer #1

(Remarks to the Author)

In this work, the author provides supporting evidence for the "error catastrophe" theory of aging, which posits that the constant and exponential accumulation of errors in protein synthesis is the root cause of damage accumulation in aged cells. More specifically, the author demonstrates that the relationship between translation fidelity and lifespan is only detectable by characterizing long-lived individuals, given certain constraints to which translation fidelity is bound. This work builds on existing studies that have already established a link between translation fidelity and longevity. The authors carefully cite these studies and expand on them by providing additional information regarding the detectability of this phenomenon. While the ideas proposed by the authors are interesting, some aspects of their model feel superficial and would benefit from further explanation and expansion to make a meaningful contribution to the field. I believe this study has the potential to be a valuable addition to the scientific community, but important concerns need to be addressed to strengthen its validity and impact.

One unclear point is why the expected lifespan should not follow a normal distribution centered around the expected lifespan value. Having a significant number of samples that fall outside the expected lifespan seems to contradict the model.

Additionally, regarding this point, the authors mention using a uniform distribution to extract samples for the model in Figure 1 (specifically in the green trapezoid area). However, the rationale for using this distribution is unclear. Do the authors refer to a uniform probability of death due to external causes? Alternatively, one would expect the expected lifespan to follow a normal distribution around some expected value, which could potentially invalidate the model proposed by the authors.

Related to this, in Figure 1C, do the authors refer to the "theoretical maximum lifespan" on the y-axis? The term "expected lifespan" could be interpreted as a probability distribution with some expected value, which, as far as I understand, is not what the authors are claiming.

Since these points form the foundation of the work, it is crucial for the authors to clearly explain all the assumptions they made and help the reader follow the hypotheses they propose. The lack of clear explanations makes interpreting their model challenging.

An interesting addition to the work would be measuring the relationship between translation error rates and longevity under conditions of nutrient scarcity or calorie restriction, as well as other lifespan-extending interventions, such as rapamycin. It is known that limiting energy sources like glucose and reducing protein synthesis extends yeast lifespan. The authors have already shown that their system detects positive changes in response to nutrient depletion. A valuable experiment would be to repeat the translation fidelity-lifespan measurements under some lifespan-extending conditions like caloric restriction and rapamycin treatment and measure if the specific genetic associations linked with it are different.

The authors should also place more emphasis on the biological significance of removing short-lived individuals from the sample. Are these individuals not "affected" by translation fidelity in relation to their longevity? It would be important for the authors to provide some characterization of these samples "below the curve." What characteristics make these yeasts suboptimal? Performing some experimental characterization of the short-lived samples could provide a more solid and meaningful context for the work.

It is also interesting to note that several loci associated with both translation fidelity and lifespan are located primarily on the X chromosome. How do the authors explain this association? I suggest expanding on possible explanations in the

discussion section of the manuscript.

Another point worth expanding in the discussion is the finding of vacuolar proteins associated with increased translation fidelity. Does this suggest that mis-translated proteins are degraded more efficiently in long-lived strains? The authors should discuss the link between autophagy/recycling mechanisms and the error catastrophe theory of aging.

Minor Points:

- Regarding Figure 1, the authors should improve the interpretability and readability of their model. Figure 1C, in particular, is difficult to interpret. As mentioned above, the authors should make a greater effort to help the reader understand the assumptions behind their model.
- In Figure 4, the authors should indicate the p-values for all the correlations shown.
- Figure 4, in general, could benefit from improved interpretability. The authors should consider displaying the different panels separately and more consistently. The use of varying visualization styles and crowded panels makes interpretation challenging.
- In Figure S1, it is unclear what the error bars on the bar plot represent. They appear to suggest a distribution of Spearman correlation values, but if I understand correctly, only one value is calculated for each removal of short-lived species. If these represent confidence intervals for the rho values, this should be specified in the figure legends.
- For Figure S1, Figure 1E, and Figure 4, confidence intervals (if applicable) should be clearly represented with error bars that are distinguishable from the overall background color. The use of the same color for both the bar plot and error bars makes interpretation difficult. Additionally, why are confidence intervals missing from Figure 4?

(Remarks on code availability)

Reviewer #2

(Remarks to the Author)

Review of "Translational Fidelity and Longevity are genetically linked"

Understanding the mechanisms of aging is an important challenge for modern biological research. Zheng et al. focused their effort on the Error-Catastrophe Theory of Aging proposed by Leslie Orgel in the 1960s. This theory states that the accumulation of errors that occur during mRNA translation in the components of the fidelity machinery compromises its function, leading to more errors, and eventually resulting in death. The theory makes a number of predictions, and the authors choose to focus on the correlation between translation fidelity and longevity, a relationship that has been observed when comparing across species. Using Orgel's mathematical model for translation error propagation, the authors performed simulations and found that the fidelity-longevity correlation improves when you remove short-lived individuals from a study. To empirically test this relationship within a species, the authors measured translation error rate and chronological lifespan in a panel of recombinant progeny from a cross between two strains of the model yeast *Saccharomyces cerevisiae*. The authors claim that their results strongly support a role for translational fidelity in the genetics of longevity in yeast. However, there are multiple serious statistical and experimental issues that need to be addressed before this claim is supported. The authors also draw a connection between their genetic signal for the yeast gene *VPS70* and its human ortholog *NAALAD2*, claiming that this gene is associated with longevity in the top 10% of long-lived individuals in the UK Biobank. The authors do not support this connection with rigorous statistics, weakening their claim that they discovered an evolutionary conserved link between translation error and aging.

Below are comments to authors organized by the sections in the manuscript, and further categorized as major and minor.

"The Translational Fidelity-Longevity Correlation Concealed by the Limited Variation of Translation Error Rate"

Major:

1. This section explores the relationship between longevity and translation fidelity by simulating from Orgel's model with additional constraints on the variation in translational fidelity. The authors place limits L and U on this variable, creating a trapezoid relationship, with the conclusion that focusing on long-lived individuals improves the correlation between translation fidelity and longevity as long as you don't remove too many samples. This conclusion depends on the assumptions of this model and the limits on translation fidelity. In their own yeast data, the authors observe a continuous distribution of translation error rates. Thus, the authors should explore whether and to what extent the limits L and U (and how strict they are) are necessary to reach the conclusion.

Minor:

1. In Fig 1e the authors show the improved correlation depending on the percentage of long-lived samples used. There is an implicit trade-off between power and the strength of the correlation, and the authors should comment on this relationship. In Fig 1e, the correlation is present regardless of the slice of the data used, suggesting that in their simulations one could observe this correlation in all instances. As part of this, the authors should include the statistics for the correlation in the text or figure to help the reader understand the relationship between slicing the data, power, and the strength of the correlation.

2. The authors claim the simulations are performed using realistic parameters, but I could not find additional justification for these parameters in the methods. The authors should provide these in the methods.

“Experimental assessment of chronological lifespan and translational fidelity in a panel of yeast recombinant inbred lines”

Major:

1. The authors used a recovery assay to determine the chronological lifespan of 804 out of a total of 1,051 yeast strains. In the methods the authors write that all strains displayed growth inhibition when treated with nourseothricin, and one entire plate was removed for not growing. Were these strains not growing at all, or only when treated with nourseothricin? Removing these 140 strains results in 911 strains, and then 111 are removed because of the quality of the CLS data? If this is correct the authors need to clarify the reasons for data exclusion in the methods.
2. Throughout the manuscript the authors refer to the yeast as recombinant inbred lines, but they are actually recombinant haploid progeny. This should be changed in the text.
3. The authors recovered transformants with a translational reporter in 260 of the 804 (out of 1051 total) strains. Were only 260 strains transformed or were some lost during construction of the reporter strains? The authors should explain why 260 strains were used and how this subset was arrived at. The parental diploid strains reported in references 52 and 53 have other MX cassettes, which will be segregating in the progeny, and could interfere with the integration of another drug marker due to shared homology. In fact, the parental strain contains a NatMX cassette replacing the FLO8 gene, suggesting that ~50% of the progeny should grow in nourseothricin. Do the authors have an explanation for why their strains showed growth inhibition on nourseothricin?

Minor:

1. When reporting statistics in the text it would be helpful to include the test used to generate the p-value. For example (F=33.20, P <10⁻¹⁵) on lines 184 and 185.
2. The authors should include confidence intervals on their reported correlation values and effects.
3. The standard errors reported for the narrow-sense and broad-sense heritability are not reported on the same scale as the estimate, which is in percentages. For example on line 189 the heritability is 33.08%, but the S.E is reported as 0.0568 when it should instead be reported as 5.68%.

“Translational fidelity is correlated with longevity among long-lived strains”

Major:

1. The authors observed no overall correlation between CLS and translational fidelity; however, when analyzing the most long-lived strains the authors observed a significant correlation. Were these correlations adjusted for multiple testing? This conclusion is a key connection between simulations from Orgel’s model and the experimental data. As it stands, the text overstates the strength of this connection.
2. The parental strain RM has higher translational error rate and higher CLS and violates the relationship proposed by Orgel. This suggests that there exist a number of loci in RM that increase translation error rate and CLS independently or together. Do the authors have an explanation for how this fits into their model?

Minor:

1. The relationship between CLS and translational fidelity could be confounded with growth rate, i.e. strains with slower growth rate live longer because of superior resource management. The authors should discuss the relationship between these variables, and its impact on the conclusions presented in the manuscript.

“The gene VPS70 underlie the correlated variation of longevity and translational fidelity”

Major:

1. The authors performed QTL mapping in 118 long lived strains that were measured for both CLS and translational error rate, and showed a negative correlation between them. There is no expectation that the QTL map should change because they are focusing on this set of strains. In the discussion, the authors mention Figure S5, in which the authors show the results of QTL mapping on every strain with measured phenotype data (N=804 for CLS, N=260 for translational error rate). For their extended CLS dataset they found the same locus on chromosome X surrounding VPS70, but for the translational error rate they do not observe a significant signal on chromosome X. The explanation given in the discussion is that the correlation is confounded by loci that do not adhere to Orgel’s rule. The analyses presented in S5F and S5G are meant to show that if you adjust for these loci you can recover the correlation. While these results show that if you remove strains segregating alleles that violate Orgel’s rule from the data you can recover a negative correlation, the authors do not recover a significant association for translational error on chromosome X. These results suggest that VPS70 is not significantly associated with translational error rate. Nevertheless, their allele replacement experiments do confirm that the VPS70 allele of RM increases CLS and decreases translational error rate. The authors should update the manuscript to make clear that the evidence for the QTL on chromosome X is marginal (possibly due to the small sample size of the translational error rate

QTL mapping).

“NAALAD2, the human ortholog of VPS70, contains a longevity-associated SNP rs10830430.”

Major:

1. The connection between the VPS70 results in yeast and humans is not well supported. Overall they found no association between NAALAD2 or FOLH1 and lifespan in the UK biobank dataset, which is unsurprising since only 1,160 samples with available lifespan data could be used to test for association. The authors then analyzed only the top 10% of these individuals and found one association. According to Figure 6, this single variant increases the average lifespan of these individuals by 2 years from 76 to 78, an incredibly large effect for a common variant in the human population discovered in a very small sample. These results suggest that the association between rs10830430 and lifespan is spurious, suggesting that an unmodelled confounding factor explains the significant association.
2. The authors make no mention of adjusting for population structure in their GWAS analysis using fixed-effect principal components (PCs) or a linear mixed model (LMM). Similarly, other confounding factors, such as sex, could explain why they observed an association.
3. The life expectancy in the UK is 80 years of age, showing that the individuals used in the UK biobank analyses are not in the top 10% of aged individuals, where under Orgel's theoretical framework we might expect to observe a correlation between lifespan and translational error rate.

(Remarks on code availability)

Reviewer #3

(Remarks to the Author)

(Remarks on code availability)

The code provided at the github link is missing a README, but since it only recreates the figure the code is quite straightforward. The figures are reproducible from the code, all other analyses are not. The repository should be expanded to include the scripts used to perform genetic mapping, CLS determination, translational fidelity analysis, and the results of other analyses presented in the manuscript.

Version 1:

Reviewer comments:

Reviewer #2

(Remarks to the Author)

The authors have addressed my previous comments point by point and made numerous revisions to the manuscript. However, I have a few remaining serious concerns outlined below.

I thank the authors for including Supplemental Table S1, which describes the reasoning for whether a strain was included in each analysis. Can the authors elaborate more on why 441 of the 701 strains with the dual luciferase reporter had unreliable luminescent estimates? Given the large number of strains excluded, did the authors investigate whether the quality of the reporter measurements is heritable?

I also remain concerned about the reported human association with SNP rs80078229 in NAALAD2. The association only appears significant when considering a subset of SNPs in the homologs of VPS70, rather than in a genome-wide analysis. Moreover, this SNP differs from the one reported in the original submission (rs10830430), raising concerns about the consistency of the findings. The fact that the associated SNPs vary depending on the chosen lifespan thresholds and statistical methodologies further argues that these associations are spurious. Given these concerns about rigor and reproducibility, I strongly recommend that the authors remove this results section from the final manuscript.

(Remarks on code availability)

Reviewer #3

(Remarks to the Author)

(Remarks on code availability)

Reviewer #5

(Remarks to the Author)

Comment 1:

The authors have provided convincing information both in the rebuttal letter and in the revised manuscript on why there might be a significant number of samples that fall outside the expected lifespan (and which could seem to contradict the aging model the authors propose). In these, cellular factors such as genomic instability and mitochondrial dysfunction might strongly contribute to the aging process. The telomere attrition that the authors mention might be related to age-related changes in telomerase activity.

Comment 2:

The authors state in their rebuttal that:

“We have conducted additional experiments to determine the translational error rate and chronological lifespan of the key strains under rapamycin treatment, which lengthens lifespan and enhances translational fidelity through affecting the mTOR pathway^{3,4}”.

Overall, the data presented by the authors is reasonably supportive, however, I am a bit worried that rapamycin treatment does not significantly influence the chronological lifespan of the BY cells (please compare DMSO (mock) and rapamycin (treatment), upper panels, BY cells, Fig. S6). However, there is a change on translational efficiency (lower panels). How do the authors explain the apparent lack of (anti) aging effect of rapamycin in the BY cells?

Comment 3:

The additional experiments conducted by the authors and their conclusions outlined in the discussion section answers this reviewer's comment reasonably well.

Comments 4 and 5:

The section in the discussion that the authors added to their manuscript is a plausible rebuttal of the comments 4 and 5 raised by the reviewer.

Minor comment 1-4:

All the comments have been addressed well by the authors.

(Remarks on code availability)

Reviewer #6

(Remarks to the Author)

I joined the revision process after the first round and was asked to focus on the authors' responses to the Reviewer1 #'s comments. I mainly followed this request, but I also added two questions/caveats to which I would appreciate a response from the authors. I read the manuscript with interest and believe the work is valuable. In my opinion, the authors addressed the majority of Reviewer #1's comments, but some ambiguities remain. I'm providing the details below.

Comment1

The authors explained that, in their experiment, the measured chronological lifespan predominantly reflects the contribution of genetic factors. They further clarified that they assume the actual lifespan of an individual strain is determined by the most severe factor. While I understand this reasoning, I believe that genetic factors may have an additive effect on lifespan, which would generally lead to a normal distribution. Furthermore, the authors' empirical data on chronological lifespan and translation error rates follow a normal distribution, not a uniform distribution.

I am unsure if the assumption about the shape of the maximal lifespan distribution is crucial to the proposed model.

However, I think it might be worthwhile to test this assumption by rerunning the simulations with the lifespan and baseline translation error rate values of the strains sampled from a normal distribution.

The authors have changed the labels in Figure 1C as suggested.

I think most of the model's assumptions have been explained. One point that is unclear to me is whether the obtained results depend on the assumption that the time needed for error catastrophe exceeds the lifespan of most individuals. This would constrain the combination of the E, D, and alpha parameters.

Comment2

The authors performed an additional experiment involving rapamycin treatment of strains with VPS70-BY and VPS70-RM alleles. They observed that the RM allele's effect was eliminated in the presence of rapamycin. I think this experiment is a sufficient addition. However, I have a question: Why is the expected effect of rapamycin treatment—lengthening the lifespan

and decreasing the error rate—not visible in Figure S6? In fact, the error rate seems elevated.

Comment3

The authors performed additional tests using a very interesting approach. They selected the test samples in a way that diminished the influence of loci that affect lifespan but not the error rate. I think the analyses are satisfactory. The experiments made me ask an additional question: What would happen if correlation tests were performed on segregants with alleles that lead to a decreased lifespan? I believe the proposed model predicts a lack of correlation.

Comment4 and 5

The authors expanded the discussion as requested.

Reading the answer caused me to speculate that the effect of the VSP70 allele on translational fidelity might be an artifact of the luciferase assay. The method does not measure the error rate directly, but rather the concentration of erroneously translated firefly luciferase, which can potentially be affected by protein sorting and degradation. Could the authors comment on this?

All minor comments have been addressed.

I used the deepL Write tool to correct the English in my comments.

(Remarks on code availability)

Version 2:

Reviewer comments:

Reviewer #2

(Remarks to the Author)

The authors have addressed my previous comments. We especially thank the authors for removing the section about the human association with SNP rs80078229. My major concerns with this manuscript have been addressed.

(Remarks on code availability)

Reviewer #3

(Remarks to the Author)

(Remarks on code availability)

Reviewer #5

(Remarks to the Author)

My comment has been adequately addressed by the authors. As such, I recommend publication of the manuscript in Nature Communications.

(Remarks on code availability)

Reviewer #6

(Remarks to the Author)

Thank you for the additional analyses and clarifications.

Comment 1:

I'm satisfied with the response and the additional simulations.

Comment 2:

Thank you for the clarification.

Comment 3:

Thank you for the analyses. I'm not sure what short-living segregants were used in them. My question was specifically about whether the correlation between lifespan and error rate will be visible for segregants having the BY alleles on ChrXII:660,371 and on ChrXIV:461,485. However, I admit that I might not have explained it sufficiently, so I don't expect any additional analyses here.

I think the issue should be discussed directly in the manuscript fragments that refer to the effects of the VPS70 alleles.

I used DeepL Write for English correction.

(Remarks on code availability)

Point-by-point response to reviewers' comments

Reviewer #1:

Overall Comment:

In this work, the author provides supporting evidence for the "error catastrophe" theory of aging, which posits that the constant and exponential accumulation of errors in protein synthesis is the root cause of damage accumulation in aged cells. More specifically, the author demonstrates that the relationship between translation fidelity and lifespan is only detectable by characterizing long-lived individuals, given certain constraints to which translation fidelity is bound. This work builds on existing studies that have already established a link between translation fidelity and longevity. The authors carefully cite these studies and expand on them by providing additional information regarding the detectability of this phenomenon. While the ideas proposed by the authors are interesting, some aspects of their model feel superficial and would benefit from further explanation and expansion to make a meaningful contribution to the field. I believe this study has the potential to be a valuable addition to the scientific community, but important concerns need to be addressed to strengthen its validity and impact.

Response:

Our sincere thanks go out to the reviewer for his/her very positive comments and constructive suggestions.

Comment 1:

One unclear point is why the expected lifespan should not follow a normal distribution centered around the expected lifespan value. Having a significant number of samples that fall outside the expected lifespan seems to contradict the model.

Additionally, regarding this point, the authors mention using a uniform distribution to extract samples for the model in Figure 1 (specifically in the green trapezoid area). However, the rationale for using this distribution is unclear. Do the authors refer to a uniform probability of death due to external causes? Alternatively, one would expect the expected lifespan to follow a normal distribution around some expected value,

which could potentially invalidate the model proposed by the authors.

Related to this, in Figure 1C, do the authors refer to the "theoretical maximum lifespan" on the y-axis? The term "expected lifespan" could be interpreted as a probability distribution with some expected value, which, as far as I understand, is not what the authors are claiming.

Since these points form the foundation of the work, it is crucial for the authors to clearly explain all the assumptions they made and help the reader follow the hypotheses they propose. The lack of clear explanations makes interpreting their model challenging.

Response:

Thank you very much for pointing out these logical ambiguities. We would like to address these ambiguities from several perspectives. **First**, we agree that the term "theoretical maximum lifespan" better reflects our intended reasoning. More specifically, we proposed that, while the expected lifespan can be affected by a number of genetic factors, it cannot exceed the maximum lifespan determined by the accumulation of translation errors (the Error-Catastrophe Theory of Aging). Therefore, we have changed all related terms in the texts, and in Figure 1 (pasted below).

Second, the rationale behind the uniform distribution in the green trapezoid area is

that the expected lifespan is influenced by other genetic factors. This can be best explained by referring to our experiments in yeast, where the chronological lifespan of each strain is essentially measured as an average of millions of cells in the same controlled laboratory environment. As such, it is likely that the influence of deaths caused by external/environmental factors will have a negligible effect across different strains. Instead, we propose that before reaching the maximum lifespan dictated by the Error-Catastrophe, deaths due to other genetic mechanisms might occur. Some of these may include telomere attrition, genomic instability, mitochondrial dysfunction, etc.

Last but not least, we would like to clarify our underlying assumption more intuitively as the “buckets effect” – the maximum practical capacity of a bucket with staves of unequal length is limited by the length of the shortest staff. Similarly, when multiple genetic factors gives different maximum lifespans, the ultimate expected lifespan is dictated by the one factor with the shortest maximum lifespan. Therefore, the effect of the Error-Catastrophe on aging is much better revealed when it is likely the “shortest staff”. We hope the reviewer is satisfied with this clarification. We have now explicitly described this in the main text, as pasted below.

To examine the intraspecific correlation between translational fidelity and longevity, we revisited Orgel's mathematical model for translation error propagation^{1,2} (**Figure 1a**). With e_t denoting the aggregate translation error rate at time t , and E denoting the baseline translation error rate, Orgel¹ proposed that $e_{t+1} = E + \alpha e_t$, where α is the proportionality constant between errors in the synthetic apparatus built at the previous timepoint and errors in proteins that is newly synthesized (in the next timepoint). The error catastrophe occurs if $\alpha \geq 1$ (REF¹, but see **Discussions**), since translation error rate (e_t . See y axis of **Figure 1a**), and consequently the mortality risk (gray scale in **Figure 1a**), increases indefinitely with t . Without loss of generality, we assumed a critical level of mortality risk with an aggregate error rate of D , so that an theoretical maximum lifespan could be expressed as the time taken for e to increase from $e_0 (= E)$ to D (“Lifespan” in **Figure 1a**, see **Methods**). Obviously, in a population, individuals/samples with a higher or lower baseline error rate E will respectively have shorter or longer maximum lifespan (**Figure 1a/b**, red and blue symbols, respectively). Thus, if the maximum lifespan were not affected by factors other than the translation error, we should predict a perfect anticorrelation between maximum lifespan and the basal translation error rate as shown by the lifespan-to-error-rate curve in **Figure 1b**. There may, however, be deaths due to other causes before the maximum lifespan that translation error dictates. Therefore, our model predicts a maximum expected lifespan based on the translation error rate, or, in other words, that lifespan of different samples will fall below the lifespan-to-error-rate curve in **Figure 1b**. This can also be explained more intuitively

as the “buckets effect” - the maximum practical capacity of a bucket with staves of unequal length is limited by the length of the shortest stave. In the same way, the samples below the lifespan-to-error-rate curve in **Figure 1b** represent deaths caused by genetic factors other than the Error-Catastrophe, such as telomere attrition, genomic instability, mitochondrial dysfunction, etc.

Comment 2:

An interesting addition to the work would be measuring the relationship between translation error rates and longevity under conditions of nutrient scarcity or calorie restriction, as well as other lifespan-extending interventions, such as rapamycin. It is known that limiting energy sources like glucose and reducing protein synthesis extends yeast lifespan. The authors have already shown that their system detects positive changes in response to nutrient depletion. A valuable experiment would be to repeat the translation fidelity-lifespan measurements under some lifespan-extending conditions like caloric restriction and rapamycin treatment and measure if the specific genetic associations linked with it are different.

Response:

Thank you very much for your highly valuable and constructive suggestion. We have conducted additional experiments to determine the translational error rate and chronological lifespan of the key strains under rapamycin treatment, which lengthens lifespan and enhances translational fidelity through affecting the mTOR pathway^{3,4}. We observed that the effect of the RM allele of VPS70 (VPS70-RM) is eliminated with rapamycin treatment, suggesting that function of VPS70 is dependent on the mTOR pathway as well. **Figure S6** now incorporated these results, which is pasted below along with its figure legend.

Figure S6. Additional results on the function of *VPS70*

Similar to Figure 5g, but with results from an independent set of experiments comparing DMSO and rapamycin treatments.

Comment 3:

The authors should also place more emphasis on the biological significance of removing short-lived individuals from the sample. Are these individuals not "affected" by translation fidelity in relation to their longevity? It would be important for the authors to provide some characterization of these samples "below the curve." What characteristics make these yeasts suboptimal? Performing some experimental characterization of the short-lived samples could provide a more solid and meaningful context for the work.

Response:

Thank you very much for this insightful question. We have performed additional QTL analyses in order to examine the characteristics of these yeast strains "below the curve". We observed in the QTL mapping results of chronological lifespan that when we

controlled the second (choosing strains with Thymine on chrXII:660,371) and third (choosing strains with Guanine on chrXIV:461,485) most significant QTLs, the correlation between translational fidelity and chronological lifespan became significant, revealing the effect of Error-Catastrophe on chronological lifespan.

We then examined these two QTLs that conceal the effect of Error-Catastrophe in greater detail. The chrXIV:461,485 position lies within the ORF YNL088W, which encodes Topoisomerase II, whose main function is to relieve torsional strain in DNA by cleaving and re-sealing phosphodiester backbones of both positively and negatively supercoiled DNA. The chrXII:660,371 position does not belong to any ORF. However a 10kbp region surrounding it contains several known genes including YLR257W (encoding a protein of unknown function), YLR258W (a glycogen synthase), YLR259C (tetradecameric mitochondrial chaperonin). We suspect that the other alleles of these two loci (Cytosine on chrXII:660,371 and Adenine on chrXIV:461,485) may limit yeast lifespan to a point before translation errors accumulate to lethality (i.e. maximum lifespan dictated by the Error-Catastrophe) through either altering genomic stability (the function of Topoisomerase II) or proteostasis (the function of chaperonin). This is, however, highly speculative. After much contemplation, we decided to limit ourselves to describing the empirical observations made by controlling the two loci and naming the surrounding genes. This part is included in the Discussion section, which is pasted below.

... when we focused our analyses on strains with Thymine on chrXII:660,371 (the RM allele associated with longer lifespan) and Guanine on chrXIV:461,485 (the RM allele associated with longer lifespan), longevity and translational fidelity exhibit significant correlations (**Supplemental Figure S7e**) as well as overlapping QTL peaks (**Supplemental Figure S7f**), although both peaks fail to achieve statistical significance. These two loci were examined more closely and it was discovered that the chrXIV:461,485 position is located within the ORF YNL088W, which encodes Topoisomerase II, a protein that relieves torsional strain in DNA by cleaving and resealing phosphodiester backbones of both positively and negatively supercoiled DNA. The chrXII:660,371 position does not correspond to any ORF, but a 10kbp region surrounding it contains several known genes, including YLR257W (encoding a protein of unknown function), YLR258W (a glycogen synthase), and YLR259C (tetradecameric mitochondrial chaperonin). The development of more general statistical methods for the systematic detection of similarly obscured phenotype-phenotype correlations or genotype-phenotype associations (or cryptic QTL) might be worthwhile in the future, such as the most recent effort on the level of a chromosome by epic-QTL⁵.

Comment 4 and 5:

It is also interesting to note that several loci associated with both translation fidelity and lifespan are located primarily on the X chromosome. How do the authors explain this association? I suggest expanding on possible explanations in the discussion section of the manuscript.

Another point worth expanding in the discussion is the finding of vacuolar proteins associated with increased translation fidelity. Does this suggest that mis-translated proteins are degraded more efficiently in long-lived strains? The authors should discuss the link between autophagy/recycling mechanisms and the error catastrophe theory of aging.

Response:

Thank you very much for these two critical questions. Since they are highly related, we will respond together. The observation that the loci associated with both translational fidelity and lifespan are both located primarily on chromosome X suggests that, while further expanding the number of strains might allow identification of other loci associated with either translation fidelity and/or lifespan, at least in this (sub)population of the strains, the strongest genetic factors underlying the variation of translational fidelity and lifespan are either identical or at least closely linked on the same chromosome. As for the function of vacuolar protein sorting 70 (VPS70), we indeed suspect that it regulates the selective sorting of mistranslated proteins into the vacuole, and/or the efficacy of vacuoles in degrading erroneous proteins. As a consequence, if the vacuolar function is blocked, the difference between alleles of VPS70 will be diminished or eliminated. Indeed, this was observed in our experimental treatment of BY and BY::VPS70-RM strains with ConA, an inhibitor of vacuolar function. As pasted below, we have expanded this in Discussion.

With genome-wide QTL mapping in the long-lived strains, we found the loci with strong associations with both translation fidelity and lifespan on chromosome X, suggesting that the genetic factors linking the two traits are primarily located on this chromosome. Our subsequent experiments tested each candidate gene individually and narrowed down the key factor to *VPS70*, a gene known to be involved in vacuolar protein sorting. From a mechanistic perspective, it has previously been demonstrated that vacuoles are functionally linked to biological processes that recycle damaged/erroneous proteins⁶, such as autophagy, a known anti-aging mechanism^{7,8}. Changes in vacuolar pH also affect lifespans in yeast^{6,9}. Thus, our identification of *VPS70* simultaneously impacts translational fidelity and longevity suggests that mistranslated proteins may be degraded more efficiently in long-lived strains via a vacuole-dependent process, which is indeed supported by our

preliminary experiments (Figure 5g) with ConA, an inhibitor of protein degradation in the vacuole¹⁰⁻¹².

Minor Comment 1:

- Regarding Figure 1, the authors should improve the interpretability and readability of their model. Figure 1C, in particular, is difficult to interpret. As mentioned above, the authors should make a greater effort to help the reader understand the assumptions behind their model.

Response:

Thank you again for highlighting this. Please see our response to your Major Comment 1 above.

Minor Comment 2:

- In Figure 4, the authors should indicate the p-values for all the correlations shown.
- Figure 4, in general, could benefit from improved interpretability. The authors should consider displaying the different panels separately and more consistently. The use of varying visualization styles and crowded panels makes interpretation challenging.

Response:

Thank you for these suggestions. We have now revised Figure 4 as suggested. The P value for each correlation is shown in the bottom half of Figure 4a, which follows the same visual style as Figure 1e. The scatter plots for the individual correlations have been plotted separately as Figure 4b. The revised Figure 4 is pasted below.

Minor Comment 3:

- In Figure S1, it is unclear what the error bars on the bar plot represent. They appear to suggest a distribution of Spearman correlation values, but if I understand correctly, only one value is calculated for each removal of short-lived species. If these represent confidence intervals for the rho values, this should be specified in the figure legends.

Response:

Thank you for this question. Because we have conducted 1,000 rounds of simulations, each of which gives a Spearman's Correlation Coefficient, we can therefore have a standard deviation of these 1,000 values. We have now explicitly explained this in the figure legend of Figure 1e and Figure S1, by the text pasted below.

Error bars represent standard deviation among simulations.

Minor Comment 4:

- For Figure S1, Figure 1E, and Figure 4, confidence intervals (if applicable) should be

clearly represented with error bars that are distinguishable from the overall background color. The use of the same color for both the bar plot and error bars makes interpretation difficult. Additionally, why are confidence intervals missing from Figure 4?

Response:

Thank you for these questions. Per your suggestion, we have revised these figures to show the error bars in a color that can be distinguished from the bar plot. Additionally, We have included confidence internals in Figure 4, which is based on 1,000 bootstraps of the real data. These revised figures are pasted below.

Reviewer #2:

Overall Comments:

Understanding the mechanisms of aging is an important challenge for modern biological research. Zheng et al. focused their effort on the Error-Catastrophe Theory of Aging proposed by Leslie Orgel in the 1960s. This theory states that the accumulation

of errors that occur during mRNA translation in the components of the fidelity machinery compromises its function, leading to more errors, and eventually resulting in death. The theory makes a number of predictions, and the authors choose to focus on the correlation between translation fidelity and longevity, a relationship that has been observed when comparing across species. Using Orgel's mathematical model for translation error propagation, the authors performed simulations and found that the fidelity-longevity correlation improves when you remove short-lived individuals from a study. To empirically test this relationship within a species, the authors measured translation error rate and chronological lifespan in a panel of recombinant progeny from a cross between two strains of the model yeast *Saccharomyces cerevisiae*. The authors claim that their results strongly support a role for translational fidelity in the genetics of longevity in yeast. However, there are multiple serious statistical and experimental issues that need to be addressed before this claim is supported. The authors also draw a connection between their genetic signal for the yeast gene *VPS70* and its human ortholog *NAALAD2*, claiming that this gene is associated with longevity in the top 10% of long-lived individuals in the UK Biobank. The authors do not support this connection with rigorous statistics, weakening their claim that they discovered an evolutionary conserved link between translation error and aging.

Below are comments to authors organized by the sections in the manuscript, and further categorized as major and minor.

Response:

Thank you very much for acknowledging the importance of our study. Please find below our point-by-point response to your comments.

Comment 1 - on “The Translational Fidelity-Longevity Correlation Concealed by the Limited Variation of Translation Error Rate”

Comment 1 – Major 1

This section explores the relationship between longevity and translation fidelity by simulating from Orgel's model with additional constraints on the variation in translational fidelity. The authors place limits L and U on this variable, creating a trapezoid relationship, with the conclusion that focusing on long-lived individuals improves the correlation between translation fidelity and longevity as long as you don't remove too many samples. This conclusion depends on the assumptions of this model and the limits on translation fidelity. In their own yeast data, the authors observe a continuous distribution of translation error rates. Thus, the authors should explore whether and to what extent the limits L and U (and how strict they are) are necessary

to reach the conclusion.

Response:

Thank you for your critical question. We would like to respond from four different perspectives. First, we agree that our yeast data show a continuous distribution of translation error rates with no obvious upper or lower limits (U and L). This can be explained, however, by the fact that all experimental measurements are inherently subject to technical noise or measurement inaccuracy. In the presence of technical noise, even genotypes with a actual translation error rates at the upper limit of $U=10^{-3}$ could have a chance to be empirically measured as $> U$. The magnitude of such chance of have an empirical translation error exceeding U depends on the statistical distribution of the technical noise. The same logic applies to the lower limit L. Therefore, the distribution of the empirically measured translation error rates does not contradict the assumption that L and U exist.

Second, the translation error rate was observed to be far outside the empirical distribution in **Figure 3d** under various (unnatural) treatments. For example, on the upper limit, an earlier study (PMID: 33758186) also showed an increase in translation error rates beyond $U=10^{-3}$ when aminoglycoside was administered to samples. On the lower limit, in our additional experiments in this revision using life-expanding treatments (rapamycin), we observed translation error rates as low as 4.96×10^{-4} , which is lower than 99.6% (259/260) of the recombinant haploid progeny strains. These results suggest that the natural variation of translation error rates observed among the yeast strains is strongly constrained within a very narrow range.

Third, we would like to emphasize again the large body of previous studies pertaining to the fitness consequences of changes in translational error rates. For example, on the U side (upper limit), it has been shown that an increase in translational error rate can result in serious problem of mistranslation-induced protein misfolding and consequent cytotoxicity¹³⁻¹⁵. On the L side (lower limit), various studies have shown that a certain degree of mistranslation is necessary for survival in stressful environments¹⁶, enhances evolvability, and facilitates adaptive evolution¹⁷⁻¹⁹. The assumption of a limited variation in translation error rates is quite reasonable based on these prior studies.

Finally, we examined the effect of varying L and U on our model in our simulation (**Figure S1**, the two left-most columns). We found that varying U and L by threefold did not alter our observation of enhanced signals for the Error-Catastrophe Theory of Aging among long-lived strains. Based on these four lines of reasoning, we conclude that a very limited range defined by L and U is likely true and that our model is robust to reasonable changes in L and U values.

Comment 1 – Minor 1

1. In Fig 1e the authors show the improved correlation depending on the percentage of

long-lived samples used. There is an implicit trade-off between power and the strength of the correlation, and the authors should comment on this relationship. In Fig 1e, the correlation is present regardless of the slice of the data used, suggesting that in their simulations one could observe this correlation in all instances. As part of this, the authors should include the statistics for the correlation in the text or figure to help the reader understand the relationship between slicing the data, power, and the strength of the correlation.

Response:

Thank you for highlighting this point. We have mentioned the trade-off between the power and the strength of the correlation when we first explain the expectation, which is pasted below:

... the fidelity-longevity correlation should increase gradually as we remove short-lived individuals from the study (**Figure 1d**), until the sample size becomes too small for meaningful statistical analysis.

Additionally, we have shown the statistical significance of the correlation (P value of Pearson's Correlation Coefficient) for the simulated dataset (**Figure 1e** and **Figure S1**) and the empirical dataset (**Figure 4**) in the lower panel of all their corresponding figures. Some of them (for instance, the panel at the top-right corner of **Figure S1**) clearly showed a decrease in statistical significance when the threshold for long-lived samples becomes too stringent, leaving too few samples to be analyzed. For your convenience, we have pasted the related figures below.

- **Figure 1e**

- Figure S1

- Figure 4

Comment 1 – Minor 2

2. The authors claim the simulations are performed using realistic parameters, but I could not find additional justification for these parameters in the methods. The authors should provide these in the methods.

Response:

Thank you for highlighting this. As pasted below, the Methods now explicitly describe the justification for these parameters.

Specifically, α is selected as a conservative value fulfilling Orgel's original proposition ($\alpha > 1$)^{1,20}; U and L are derived from previous luciferase-based²¹ and mass-spectrometry-based²² measurements of translation error rates in yeast; D is chosen to be slightly higher than the highest translation error rate observed in microbes treated with rifampicin²³. To explore potential inaccuracies of the above parameters and test the robustness of the observed pattern, each parameter was individually varied two to nine-fold.

Comment 2 - on “Experimental assessment of chronological lifespan and translational fidelity in a panel of yeast recombinant inbred lines”

Comment 2 – Major 1

The authors used a recovery assay to determine the chronological lifespan of 804 out of a total of 1,051 yeast strains. In the methods the authors write that all strains displayed growth inhibition when treated with nourseothricin, and one entire plate was removed for not growing. Were these strains not growing at all, or only when treated with nourseothricin? Removing these 140 strains results in 911 strains, and then 111 are removed because of the quality of the CLS data? If this is correct the authors need to clarify the reasons for data exclusion in the methods.

Response:

Thank you very much for these questions. Your understanding is correct. To clarify, we have added an explicit listing of the strains included/excluded by each quality filter in a new **Supplemental Table S1**. The entire plate that was removed is not growing when treated with nourseothricin. In the original study, this plate was designated "A9", whose inconsistent growth status can be seen in the newly added **Supplemental Table S1**.

Comment 2 – Major 2

Throughout the manuscript the authors refer to the yeast as recombinant inbred lines, but they are actually recombinant haploid progeny. This should be changed in the text.

Response:

Thank you for the very important point. We have revised this across our manuscript.

Comment 2 – Major 3

The authors recovered transformants with a translational reporter in 260 of the 804 (out of 1051 total) strains. Were only 260 strains transformed or were some lost during construction of the reporter strains? The authors should explain why 260 strains were used and how this subset was arrived at. The parental diploid strains reported in references 52 and 53 have other MX cassettes, which will be segregating in the progeny, and could interfere with the integration of another drug marker due to shared homology. In fact, the parental strain contains a NatMX cassette replacing the FLO8 gene,

suggesting that ~50% of the progeny should grow in nourseothricin. Do the authors have an explanation for why their strains showed growth inhibition on nourseothricin?

Response:

Thank you very much for these questions. Using PCR, we have validated the integration of the dual luciferase reporter system into HO, confirming that it is not interfered with by the other MX cassettes. We indeed observed that approximately half of the strains grow on nourseothricin medium (previous description in **Method** was a mistake). To clarify, we have added an explicit listing of the strains included/excluded by each quality filter in a new **Supplemental Table S1**. More specifically, among the total 1051 strains, 140 showed NAT resistance inconsistent with original report (911 strains remained); 107 were removed for lack of reliable chronological lifespan measurement (804 strains remained); 103 failed in transformation of dual luciferase reporter system (701 strains remained); 441 were removed due to lack of reliable measurements in Renilla relative luminescence units (RLUs), Firefly (mutant or wildtype) RLUs, or translation error rates (260 strains remained).

Comment 2 – Minor 1

When reporting statistics in the text it would be helpful to include the test used to generate the p-value. For example ($F=33.20$, $P < 10^{-15}$) on lines 184 and 185.

Response:

Thank you for this suggestion. We have now explicitly described that this P value is from F-test. Other mentions of P values were similarly revised.

Comment 2 – Minor 2

The authors should include confidence intervals on their reported correlation values and effects.

Response:

Thank you. This has now been revised as suggested.

Comment 2 – Minor 3

The standard errors reported for the narrow-sense and broad-sense heritability are not reported on the same scale as the estimate, which is in percentages. For example on line

189 the heritability is 33.08%, but the S.E is reported as 0.0568 when it should instead be reported as 5.68%.

Response:

Thank you. This has now been revised as suggested and pasted below.

Similarly, the fraction of lifespan variance explained by the additive effects of all segregating markers (narrow-sense heritability, or h^2) was estimated as 33.08% (S.E. = 5.68%), which sets an upper bound for the total amount of additive genetic variance that could be explained with a QTL-based model.

Comment 3 - on “Translational fidelity is correlated with longevity among long-lived strains”

Comment 3 - Major 1

The authors observed no overall correlation between CLS and translational fidelity; however, when analyzing the most long-lived strains the authors observed a significant correlation. Were these correlations adjusted for multiple testing? This conclusion is a key connection between simulations from Orgel’s model and the experimental data. As it stands, the text overstates the strength of this connection.

Response:

Thank you for this question. The significance levels we previously presented have not been adjusted for multiple testing. Per your suggestion, we now used the Bonferroni-adjusted P values. Note that the adjusted P value remained significant ($P < 0.05$) for the fidelity-longevity correlation among the long-lived strains. The accordingly revised **Figure 4** is pasted below.

Comment 3 - Major 2

The parental strain RM has higher translational error rate and higher CLS and violates the relationship proposed by Orgel. This suggests that there exist a number of loci in RM that increase translation error rate and CLS independently or together. Do the authors have an explanation for how this fits into their model?

Response:

Thank you for highlighting this very important question. We agree and have suspected that this phenomenon is due to other loci that regulate translation error rate (TER) and chronological lifespan (CLS) independently. Based on the comparison between the QTL results of the two traits (i.e. **Figure 5a**), the TER-specific peak on chromosome III appears to be an important factor, since it only affects TER but not CLS. Therefore, we extracted all genes containing nonsynonymous SNPs between RM and BY in this peak, including *ADP1*, *CTO1*, *YCR016W*, *CWH43*, *HSP30*, *YCR022C*, and *YCR023C*. In a similar manner to our previous experiments that investigated genes in the CLS-TER overlapping QTL peak on chromosome X, we used a two-step method^{24,25} to replace each of these genes in the BY parental strain with the same genes in the RM parental

strain. It is found that (see the figure below) when *CWH43* is replaced by its RM allele, the TER is significantly increased (Wilcoxon rank-sum test) by ~14.5% (median value across biological replicates, from 8.49×10^{-4} to 9.73×10^{-4}). There is currently no known function of *CWH43* that could affect TER, but we suspect that a gene involved in the cell wall synthesis might impact translational fidelity by influencing the availability of charged tRNAs, as has been demonstrated previously in bacteria²⁶. More functional studies are required to investigate this in the future. These results are now included as **Supplemental Figure S5**, which is pasted below.

Figure S5 – Identification of the translational fidelity-specific loci underlying the translation error rate of RM

The parental RM strain exhibit higher translation error rate but also longer chronological lifespan, which is opposed to the prediction of the Error-Catastrophe Theory of Ageing. This suggests the existence of other loci that regulate translation error rate and chronological lifespan independently. Based on the comparison between the QTL results of the two traits (i.e. **Figure 5a**), the TER-specific peak on chromosome III appears to be an important factor, since it only affects TER but not CLS. Therefore, we extracted all genes containing nonsynonymous SNPs between RM and BY in this peak, including *ADP1* (a putative ATP-dependent permease), *CTO1* (a protein required for cold tolerance), *YCR016W* (a RNA-binding ribosome assembly factor), *CWH43* (involved in GPI anchor biosynthesis and cell wall organization), *HSP30* (a heat shock protein), *YCR022C* (unknown function), and *YCR023C* (a vacuolar membrane protein of unknown function). We used a two-step

method^{24,25} to replace each of these genes in the BY parental strain with the same genes in the RM parental strain. It is found that when *CWH43* is replaced by its RM allele, the translation error rate is significantly increased by ~14.5% (median value across biological replicates, from 8.49×10^{-4} to 9.73×10^{-4}). This figure is similar to **Figure 5f**. *P* values from the Wilcoxon rank-sum tests that compare the BY parental strain with each of the other strains are indicated on top.

Comment 3 - Minor 1

The relationship between CLS and translational fidelity could be confounded with growth rate, i.e. strains with slower growth rate live longer because of superior resource management. The authors should discuss the relationship between these variables, and its impact on the conclusions presented in the manuscript.

Response:

Thank you for this critical point. We have now discussed this in the Discussion section, which is pasted below.

... our observations might not necessarily be explained solely by Orgel's hypothesis, as other confounding factors may be increasing translational fidelity and extending lifespan at the same time. Growth rate, for example, may be one such confounding factor, as strains with slower growth tend to live longer^{27,28} due to mechanisms such as superior resource management, which might also enhance translational fidelity. However, the Error-Catastrophe Theory of Aging still provides the most coherent explanation of the evidence that is currently available. This is because, on the one hand, the confounding by growth rate as outlined above predict a fidelity-longevity correlation regardless the strain's lifespan, meanwhile as reasoned in this study, the Error-Catastrophe Theory of Aging predicts that fidelity-longevity correlations are more apparent among the long-lived strains, which is observed in our empirical dataset. On the other hand, it is not easy to distinguish their causal relationship given the complicated interplay among translational fidelity, longevity, and growth rate. It is possible to argue, for example, that both enhanced longevity and slowed growth are the result of increased translational fidelity. More importantly, such a causal relationship is indeed directly supported by the changes in lifespan following manipulation of translational fidelity²⁹⁻³¹.

Comment 4 – on “The gene VPS70 underlie the correlated variation of longevity and translational fidelity”

Comment 4 - Major 1

The authors performed QTL mapping in 118 long lived strains that were measured for both CLS and translational error rate, and showed a negative correlation between them. There is no expectation that the QTL map should change because they are focusing on this set of strains. In the discussion, the authors mention Figure S5, in which the authors show the results of QTL mapping on every strain with measured phenotype data (N=804 for CLS, N=260 for translational error rate). For their extended CLS dataset they found the same locus on chromosome X surrounding VPS70, but for the translational error rate they do not observe a significant signal on chromosome X. The explanation given in the discussion is that the correlation is confounded by loci that do not adhere to Orgel’s rule. The analyses presented in S5F and S5G are meant to show that if you adjust for these loci you can recover the correlation. While these results show that if you remove strains segregating alleles that violate Orgel’s rule from the data you can recover a negative correlation, the authors do not recover a significant association for translational error on chromosome X. These results suggest that VPS70 is not significantly associated with translational error rate. Nevertheless, their allele replacement experiments do confirm that the VPS70 allele of RM increases CLS and decreases translational error rate. The authors should update the manuscript to make clear that the evidence for the QTL on chromosome X is marginal (possibly due to the small sample size of the translational error rate QTL mapping).

Response:

Thank you very much ! Per your suggestion, we have now added this text below when describing **Figure S5f** (now **Figure S7f**) in the Discussion, as pasted below.

...longevity and translational fidelity exhibit significant correlations (**Supplemental Figure S7e**) as well as overlapping QTL peaks (**Supplemental Figure S7f**), although both peaks fail to achieve statistical significance.

Comment 5 - “NAALAD2, the human ortholog of VPS70, contains a longevity-associated SNP rs10830430.”

Comment 5 – Major 1 / 2 / 3

The connection between the VPS70 results in yeast and humans is not well supported. Overall they found no association between NAALAD2 or FOLH1 and lifespan in the UK biobank dataset, which is unsurprising since only 1,160 samples with available lifespan data could be used to test for association. The authors then analyzed only the top 10% of these individuals and found one association. According to Figure 6, this single variant increases the average lifespan of these individuals by 2 years from 76 to 78, an incredibly large effect for a common variant in the human population discovered in a very small sample. These results suggest that the association between rs10830430 and lifespan is spurious, suggesting that an unmodelled confounding factor explains the significant association.

The authors make no mention of adjusting for population structure in their GWAS analysis using fixed-effect principal components (PCs) or a linear mixed model (LMM). Similarly, other confounding factors, such as sex, could explain why they observed an association.

The life expectancy in the UK is 80 years of age, showing that the individuals used in the UK biobank analyses are not in the top 10% of aged individuals, where under Orgel’s theoretical framework we might expect to observe a correlation between lifespan and translational error rate.

Response:

Thank you for these comments! Since they are closely related, they are addressed together here. Based on your suggestions, we have made a couple revisions to address these concerns. First, the full quality-filtered UKB dataset we analyzed includes 19,618 individuals who were sampled prior to 2021. As the top 10% longest-lived UKB samples (with a minimum lifespan of 75.2 years) are not the top 10% longest-lived individuals in the UK, following your suggestion and the data from the UK Office for National Statistics, we have chosen to use the lifespan threshold of 78.5 years instead, which is the UK’s life expectancy for males in 2021, and gives 150 individuals for statistical tests (Female life expectancy is 82.6 years in UK, which leaves no individual in the UKB dataset). We further elaborated on our reasoning in the Result sections as follows.

... Due to the absence of translation error rate data and therefore its correlation with lifespans in these samples, we are unable to follow our

methods used in yeasts to identify a proper subset of long-lived samples for testing these genes' concealed associations with lifespan. However, because our theoretical model (**Figure 1**) and yeast data (**Figure 4**) suggest focusing on the long-lived samples should generally facilitate the detection of the fidelity-longevity correlation until the sample size becomes insufficient for statistical testing, we chose to focus on the 150 samples whose lifespan exceeds the UK's life expectancy for males in 2021 (the life expectancy for females in 2021 is 82.6, which leaves no sample with known lifespan in UK Biobank). After excluding the effects of sex and population structure by the genotype-conditional association test³² (see **Methods**), we found that the SNP rs80078229 in *NAALAD2* showed a significant association with lifespan (**Figure 6b**, chi-squared test, Bonferroni adjusted $P = 0.0401$). This missense SNP corresponds to a lifespan-lengthening (one year increase of median lifespan) substitution of the ancestral Adenine to Guanine (Wilcoxon Rank Sum test $P = 0.0199$, **Figure 6c**), or Threonine to Alanine in terms of amino acids, in the 10th exon of the gene's major transcript (RefSeq:NM_005467.4, Ensembl:ENST00000534061.6). This finding, together with our other results, suggests that translational fidelity has an impact on the lifespan variation of human population.

Second, to exclude the confounding effects of population structure and sex, we resorted to the genotype-conditional association test with sex as an adjustment, which has been proved to be robust to arbitrarily complex population structure³². When we analyzed the full UKB dataset, no significant association was found. However, we found rs80078229 as significantly associated with lifespan when focusing on individuals who have a lifespan exceeding 78.5 years (Bonferroni adjusted $P = 0.0401$). There are two genotypes for this locus, AA and AG, with the latter showing a longer lifespan by one year. These revised results are shown in **Figure 6**, which is pasted below.

Figure 6. Association of FOLH1 and NAALAD2 with human longevity.

(a) Manhattan plot of association between lifespan and markers in FOLH1 (left) or NAALAD2 (right) on human chromosome 11. Gradually lighter gray backgrounds indicate progressively more significant Bonferroni adjusted P values. The gene structures of these two genes are shown at the top.

(b) Same as (a), except that only long-lived individuals with a lifespan greater than 78.5 years were analyzed. The SNP rs80078229 in NAALAD2 showed a significant association with lifespan (chi-squared test, Bonferroni adjusted $P = 0.0401$).

(c) The lifespan distribution of the two genotypes observed for rs80078229 shown by standard boxplot. The GG genotype was not found in the samples.

To conclude, our observation that there is no association in the full dataset, but that association emerges among long-lived samples is consistent with our simulation and yeast results, and supports our main argument.

Reviewer #3:

Comment 1

The code provided at the github link is missing a README, but since it only recreates the figure the code is quite straightforward. The figures are reproducible from the code, all other analyses are not. The repository should be expanded to include the scripts used to perform genetic mapping, CLS determination, translational fidelity analysis, and the results of other analyses presented in the manuscript.

Response:

The codes to reproduce our analyses, starting from the raw data to the final figures, have now been uploaded to the original GitHub repo.

1. Orgel, L.E. The maintenance of the accuracy of protein synthesis and its relevance to ageing: a correction. *Proc Natl Acad Sci U S A* **67**, 1476 (1970).
2. Witten, T.M. Modeling Cellular Aging: An Introduction – Mathematical and Computational Approaches. in *Cellular Ageing and Replicative Senescence* 117-141 (2016).
3. Haar, T.v.d. *et al.* The Control of Translational Accuracy Is a Determinant of Healthy Ageing in Yeast. *Open Biology* **7**, 160291 (2017).
4. Souza-Guerreiro, T.C.d., Meng, X., Dacheux, E., Firczuk, H. & McCarthy, J.E.G. Translational Control of Gene Expression Noise and Its Relationship to Ageing in Yeast. *Febs Journal* **288**, 2278-2293 (2020).
5. Buzby, C. *et al.* Epistasis and cryptic QTL identified using modified bulk segregant analysis of copper resistance in budding yeast. *bioRxiv* (2024).
6. Aufschnaiter, A. & Buttner, S. The vacuolar shapes of ageing: From function to morphology. *Biochim Biophys Acta Mol Cell Res* **1866**, 957-970 (2019).
7. Eisenberg, T. *et al.* Induction of Autophagy by Spermidine Promotes Longevity. *Nature Cell Biology* **11**, 1305-1314 (2009).
8. Rubinsztein, D.C., Mariño, G. & Kroemer, G. Autophagy and Aging. *Cell* **146**, 682-695 (2011).
9. Hughes, A.L. & Gottschling, D.E. An early age increase in vacuolar pH limits mitochondrial function and lifespan in yeast. *Nature* **492**, 261-5 (2012).
10. Dettmer, J., Hong-Hermesdorf, A., Stierhof, Y.-D. & Schumacher, K. Vacuolar H⁺-ATPase activity is required for endocytic and secretory trafficking in Arabidopsis. *The Plant Cell* **18**, 715-730 (2006).
11. Droese, S. *et al.* Inhibitory effect of modified bafilomycins and concanamycins on P- and V-type adenosinetriphosphatases. *Biochemistry* **32**, 3902-3906 (1993).
12. Huss, M. *et al.* Concanamycin A, the specific inhibitor of V-ATPases, binds to

- the Vo subunit c. *Journal of Biological Chemistry* **277**, 40544-40548 (2002).
13. Drummond, D.A. & Wilke, C.O. Mistranslation-induced protein misfolding as a dominant constraint on coding-sequence evolution. *Cell* **134**, 341-52 (2008).
 14. Drummond, D.A. & Wilke, C.O. The evolutionary consequences of erroneous protein synthesis. *Nat Rev Genet* **10**, 715-24 (2009).
 15. Yang, J.R., Zhuang, S.M. & Zhang, J. Impact of translational error-induced and error-free misfolding on the rate of protein evolution. *Mol Syst Biol* **6**, 421 (2010).
 16. Samhita, L., Raval, P.K. & Agashe, D. Global mistranslation increases cell survival under stress in *Escherichia coli*. *PLoS Genet* **16**, e1008654 (2020).
 17. Zheng, J., Guo, N. & Wagner, A. Mistranslation Reduces Mutation Load in Evolving Proteins through Negative Epistasis with DNA Mutations. *Mol Biol Evol* **38**, 4792-4804 (2021).
 18. Schmutzer, M. & Wagner, A. Not Quite Lost in Translation: Mistranslation Alters Adaptive Landscape Topography and the Dynamics of Evolution. *Mol Biol Evol* **40**(2023).
 19. Bratulic, S., Toll-Riera, M. & Wagner, A. Mistranslation can enhance fitness through purging of deleterious mutations. *Nat Commun* **8**, 15410 (2017).
 20. Orgel, L.E. The maintenance of the accuracy of protein synthesis and its relevance to ageing. *Proc Natl Acad Sci U S A* **49**, 517-21 (1963).
 21. Kramer, E.B., Vallabhaneni, H., Mayer, L.M. & Farabaugh, P.J. A comprehensive analysis of translational missense errors in the yeast *Saccharomyces cerevisiae*. *RNA* **16**, 1797-808 (2010).
 22. Mordret, E. *et al.* Systematic Detection of Amino Acid Substitutions in Proteomes Reveals Mechanistic Basis of Ribosome Errors and Selection for Translation Fidelity. *Mol Cell* **75**, 427-441 e5 (2019).
 23. Javid, B. *et al.* Mycobacterial mistranslation is necessary and sufficient for rifampicin phenotypic resistance. *Proc Natl Acad Sci U S A* **111**, 1132-7 (2014).
 24. Gray, M., Piccirillo, S. & Honigberg, S.M. Two-step method for constructing unmarked insertions, deletions and allele substitutions in the yeast genome. *FEMS Microbiol Lett* **248**, 31-6 (2005).
 25. Smith, E.N. & Kruglyak, L. Gene-environment interaction in yeast gene expression. *PLoS Biol* **6**, e83 (2008).
 26. Aggarwal, S.D. *et al.* A molecular link between cell wall biosynthesis, translation fidelity, and stringent response in *Streptococcus pneumoniae*. *Proc Natl Acad Sci U S A* **118**(2021).
 27. Delaney, J.R., Murakami, C.J., Olsen, B., Kennedy, B.K. & Kaeberlein, M. Quantitative evidence for early life fitness defects from 32 longevity-associated alleles in yeast. *Cell Cycle* **10**, 156-65 (2011).
 28. Longo, V.D., Shadel, G.S., Kaeberlein, M. & Kennedy, B. Replicative and chronological aging in *Saccharomyces cerevisiae*. *Cell Metab* **16**, 18-31 (2012).
 29. von der Haar, T. *et al.* The control of translational accuracy is a determinant of healthy ageing in yeast. *Open Biol* **7**(2017).
 30. Martinez-Miguel, V.E. *et al.* Increased fidelity of protein synthesis extends

- lifespan. *Cell Metab* **33**, 2288-2300 e12 (2021).
31. Shcherbakov, D. *et al.* Premature aging in mice with error-prone protein synthesis. *Sci Adv* **8**, eabl9051 (2022).
 32. Song, M., Hao, W. & Storey, J.D. Testing for genetic associations in arbitrarily structured populations. *Nat Genet* **47**, 550-4 (2015).

Point-by-point response to the comments

Response:

Thank you very much for your kind attention to our manuscript. In response to Reviewer #2's suggestion, we have now removed the section on the human gene NAALAD2. Regarding the exclusion of noisy measurements, the raw data and R codes underlying the data exclusion procedure have been included in the GitHub repository, and a further explanation of the procedure can be found in the Methods section. We emphasize that translation error rates are calculated as ratios of ratios based on four raw luciferase activities. Due to the nature of division (ratio) calculations, the noise in any single raw signal will be magnified, and therefore, the translation error rate is expected to have a high level of noise. In Supplemental Table S1, we explain, by the order of our experiments, whether a specific strain passed or failed the quality criteria of each experimental step and, thus, is included in the final list of analyzed strains. It is our hope that these efforts will enhance transparency and facilitate reproducibility of our analyses. Please find below our point-by-point response to the reviewers' comments.

Reviewer #2:

Overall Comment:

The authors have addressed my previous comments point by point and made numerous revisions to the manuscript. However, I have a few remaining serious concerns outlined below.

Response:

We would like to thank the reviewer for his/her constructive comments and acknowledgement of our previous revisions. Below are our responses to the remaining concerns.

Comment 1:

I thank the authors for including Supplemental Table S1, which describes the reasoning for whether a strain was included in each analysis. Can the authors

elaborate more on why 441 of the 701 strains with the dual luciferase reporter had unreliable luminescent estimates? Given the large number of strains excluded, did the authors investigate whether the quality of the reporter measurements is heritable?

Response:

Thank you for your question. Per your question, we conducted additional analyses. Briefly, we examined whether technical replicates with large CVs of translation error rates tended to occur repeatedly in biological replicates of the same strain. As shown in Figure RR1 below, this is indeed the case. More specifically, some strains tend to exhibit large CVs among technical replicates, whereas some other tend to exhibit low CV (Figure RR1a). Also, the CV among technical replicates is positively correlated between biological replicates of the same strain (Figure RR1b and c). This may be caused by an increased level of expression noise of tRNA or genes that interfere with the luciferase activity, or by an increased variability in cell lysis efficiency. Regardless the specific mechanism, our strict criteria of $CV < 0.2$ ensures the reliability of our conclusion.

Figure RR1. (a) The CV of translation error rate among technical replicates is shown by a color as indicated by the color scale bar on top. Some strains (y axis), such as those on top, tend to exhibit large CV among technical replicates in all three biological replicates (x axis). (b) The CVs of translation error rate among technical replicates measured from biological replicates 1 and 2 of each strain are shown. The blue line represents a linear model fitted to the data. The Pearson's correlation coefficient and *P* value are indicated. (c) The same as (b), except that results from replicates 1 and 3 are shown.

Additionally, we have revised a related paragraph (pasted below) in the Methods section. We have further clarified our criteria, and also explained this issue from a theoretical perspective towards the end of the paragraph.

The RLU data were filtered according to a stringent set of criteria in order to ensure accurate measurement of the translation error rate. We used only mutant Firefly RLUs between 5×10^2 and 10^4 , wild-type Firefly RLUs greater than 10^6 , and mutant and wild-type Renilla RLUs between 5×10^6 and 10^8 . We then calculated the protein abundance-normalized Firefly activity by dividing Firefly RLU by the Renilla RLU³⁵. Outliers or noisy measurements among the three biological replicates were filtered by the same procedure and criteria as described above in CLS measurement, at a threshold of $CV < 0.2$. A value for the biological replicate was calculated by averaging the remaining technical replicates. Then the three biological replicates were again filtered by the aforementioned procedure to remove outliers or noisy measurements before being averaged for strain-specific mutant and wild-type Firefly activities. The translation error rate of each strain was calculated as the ratio between its mutant Firefly activity and its wild-type Firefly activity³⁵. The biological replicates of mutant and wild-type Firefly were also individually used to calculate biologically replicated translation error rates, which were used to estimate the standard deviation. Finally, strains with translation error rate greater than 10^{-2} were removed, leaving 260 strains with estimated translation error rates (**Supplemental Table S1**). The many strains removed at this stage reflects three technical difficulties in measuring translation error rates. First, translation errors are extremely rare. The detection of translation error therefore requires high sensitivity, which usually comes at the expense of specificity. Second, the final translation error rate is calculated as a ratio of ratios, which magnifies measurement error (or noise). Third, we noticed that some strains tend to exhibit large CVs among technical replicates, such that the CV among technical replicates is positively correlated between biological replicates of the same strain (Pearson's $R > 0.26$, $P < 1.2 \times 10^{-5}$). This may be caused by an increased level of expression noise of tRNA or genes that

interfere with the luciferase activity, or by an increased variability in cell lysis efficiency. Due to these considerations, we implemented a strict requirement for the consistency of biological and technical replicates to ensure the validity of our conclusion, despite the omission of many samples.

It is our hope that these efforts will enhance transparency and facilitate reproducibility of our research.

Comment 2:

I also remain concerned about the reported human association with SNP rs80078229 in NAALAD2. The association only appears significant when considering a subset of SNPs in the homologs of VPS70, rather than in a genome-wide analysis. Moreover, this SNP differs from the one reported in the original submission (rs10830430), raising concerns about the consistency of the findings. The fact that the associated SNPs vary depending on the chosen lifespan thresholds and statistical methodologies further argues that these associations are spurious. Given these concerns about rigor and reproducibility, I strongly recommend that the authors remove this results section from the final manuscript.

Response:

Thank you for the question. Per your suggestion, we have now removed this section.

Reviewer #3:

Overall Comment:

Response:

Thank you very much for your suggestions.

Reviewer #5:

Comment 1:

The authors have provided convincing information both in the rebuttal letter and in the revised manuscript on why there might be a significant number of samples that fall outside the expected lifespan (and which could seem to contradict the aging model the authors propose). In these, cellular factors such as genomic instability and mitochondrial dysfunction might strongly contribute to the aging process. The telomere attrition that the authors mention might be related to age-related changes in telomerase activity.

Response:

Thank you very much for acknowledging our rebuttal. Indeed, these factors independent of translation error (mentioned in the last sentence of the Result section's first paragraph) could have led to mortality before the maximum lifespan caused by the Error-Catastrophe.

Comment 2:

The authors state in their rebuttal that:

“We have conducted additional experiments to determine the translational error rate and chronological lifespan of the key strains under rapamycin treatment, which lengthens lifespan and enhances translational fidelity through affecting the mTOR pathway^{3,4}”.

Overall, the data presented by the authors is reasonably supportive, however, I am a bit worried that rapamycin treatment does not significantly influence the chronological lifespan of the BY cells (please compare DMSO (mock) and rapamycin (treatment), upper panels, BY cells, Fig. S6). However, there is a change on translational efficiency (lower panels). How do the authors explain the apparent lack of (anti) aging effect of rapamycin in the BY cells?

Response:

Thank you for your question. In Fig.S6, the lifespan is expressed as relative to the BY parental strain. In other words, the average y value of the BY strain in both upper panels are strictly 1.0. Per your question, the absolute lifespan (area under the survival

curve, or SI) of the BY strains in DMSO and Rapamycin treatment are respectively 6.17 and 16.15. This is indeed consistent with the lifespan-extending effect of Rapamycin. To clarify, we have now explicitly stated in the figure legend that the y axes of these figures are scaled so that the average of BY is equal to 1.

Comment 3:

The additional experiments conducted by the authors and their conclusions outlined in the discussion section answers this reviewer's comment reasonably well.

Response:

Thank you very much for acknowledging our rebuttal.

Comments 4 and 5:

The section in the discussion that the authors added to their manuscript is a plausible rebuttal of the comments 4 and 5 raised by the reviewer.

Response:

Thank you.

Minor comment 1-4:

All the comments have been addressed well by the authors.

Response:

Thank you.

Reviewer #6:

Overall Comment:

I joined the revision process after the first round and was asked to focus on the authors' responses to the Reviewer1 #'s comments. I mainly followed this request, but I also added two questions/caveats to which I would appreciate a response from the

authors. I read the manuscript with interest and believe the work is valuable. In my opinion, the authors addressed the majority of Reviewer #1's comments, but some ambiguities remain. I'm providing the details below.

Comment 1

The authors explained that, in their experiment, the measured chronological lifespan predominantly reflects the contribution of genetic factors. They further clarified that they assume the actual lifespan of an individual strain is determined by the most severe factor. While I understand this reasoning, I believe that genetic factors may have an additive effect on lifespan, which would generally lead to a normal distribution. Furthermore, the authors' empirical data on chronological lifespan and translation error rates follow a normal distribution, not a uniform distribution.

I am unsure if the assumption about the shape of the maximal lifespan distribution is crucial to the proposed model. However, I think it might be worthwhile to test this assumption by rerunning the simulations with the lifespan and baseline translation error rate values of the strains sampled from a normal distribution.

The authors have changed the labels in Figure 1C as suggested.

I think most of the model's assumptions have been explained. One point that is unclear to me is whether the obtained results depend on the assumption that the time needed for error catastrophe exceeds the lifespan of most individuals. This would constrain the combination of the E, D, and alpha parameters.

Response:

Thank you. In response to your question, we attempted to change the uniform distribution-based simulation in **Figure 1e** to a normal distribution-based simulation, with all other parameters unchanged. The results are presented below in **Figure RR2**. Specifically, we simulated the translation error rates by a uniform distribution that is identical to our original setup in **Figure 1e**. And then, the maximum lifespans are first simulated via a normal distribution whose mean is 8 and standard deviation is 1.65 (chosen to make the final range similar to that in **Figure 1e**). When a simulated sample exceeds the maximum lifespan dictated by the Error-Catastrophe (given its baseline translation error rate), the sample is reduced to that theoretical maximum lifespan. In the original uniform distribution-based simulation, the quantile-based thresholds for short-lived samples are equal-spaced, which obviously would not hold in normal distribution-based simulations. To recapitulate this feature, we directly used equal-spaced thresholds for short-lived samples (*x* axis in **Figure RR2**). As a result, we can still observe the gradual increase in fidelity-longevity correlation after the progressive removal the short-lived samples.

Figure RR2. The correlation between translation error rate and longevity in computationally simulated samples. Translation error rates follow a uniform distribution. The maximum lifespans are first simulated via a normal distribution whose mean is 8 and standard deviation is 1.65. When a simulated sample exceeds the maximum lifespan dictated by the Error-Catastrophe (given its baseline translation error rate), the sample is reduced to that theoretical maximum lifespan. Spearman's Correlation Coefficient ρ and corresponding Bonferroni adjusted P values are shown in the upper and lower halves, respectively. There is a stronger correlation when a fraction (x) of the short-lived samples is removed from the analysis. Scatter plots with linear regression models in green lines are shown for three specific simulation results as insets. Red dashed lines indicate the threshold for statistical significance (i.e. Bonferroni adjusted $P = 0.05$). Error bars represent standard deviation among simulations. Key model parameters underlying these results are also listed at the top left corner.

Regarding the assumption that the time needed for Error-Catastrophe exceeds the lifespan of most individuals, we agree that this would constrain the possible combinations of parameters. However, if Error-Catastrophe occurs much earlier, this would result in many more deaths due to Error-Catastrophe. Thus, the Error-Catastrophe would have a greater impact on aging and life expectancy. As a result, our current assumption underestimates the importance of Error-Catastrophe, since only long-lived individuals are affected by the Error-Catastrophe.

Comment 2

The authors performed an additional experiment involving rapamycin treatment of strains with VPS70-BY and VPS70-RM alleles. They observed that the RM allele's effect was eliminated in the presence of rapamycin. I think this experiment is a sufficient addition. However, I have a question: Why is the expected effect of rapamycin treatment—lengthening the lifespan and decreasing the error rate—not visible in Figure S6? In fact, the error rate seems elevated.

Response:

Thank you. In Fig.S6, the lifespan is expressed as relative to the BY parental strain. In other words, the average y value of the BY strain in both upper panels is strictly 1.0. The absolute lifespan (area under the survival curve, or SI) of the BY strains in DMSO and Rapamycin treatment are respectively 6.17 and 16.15. This is indeed consistent with the lifespan-extending effect of Rapamycin.

Similarly, the translation error rate is also expressed as relative to the BY parental strain, so that the average y value of the BY strain in both bottom panels is strictly 1.0. The absolute translation error rate of the BY strains in DMSO and Rapamycin treatment are respectively 8.04×10^{-4} and 5.75×10^{-4} . This is also consistent with the error-decreasing effects of Rapamycin. To clarify, we have now explicitly stated in the figure legend that the y axes of these figures are scaled so that the average of BY is equal to 1.

Comment 3

The authors performed additional tests using a very interesting approach. They selected the test samples in a way that diminished the influence of loci that affect lifespan but not the error rate. I think the analyses are satisfactory.

The experiments made me ask an additional question: What would happen if correlation tests were performed on segregants with alleles that lead to a decreased lifespan? I believe the proposed model predicts a lack of correlation.

Response:

Thank you. In response to your question, we tried correlation tests on segregants with short lifespans. In agreement with the reviewer's prediction, and consistent with our model's prediction, we found that the fidelity-longevity correlation never reaches statistical significance (**Figure RR3** below).

Figure RR3. (a) The Spearman's Correlation Coefficient between chronological lifespan and translation error rate was calculated after a fraction (x axis, the number of strains remained is also shown) of long-lived strains was removed. The bars in the upper panel indicate Spearman's ρ , with the error bars representing the standard error of the mean assessed by 100 bootstraps, whereas the lollipops in the lower panel indicate the statistical significance, with the threshold of Bonferroni adjusted $P = 0.05$ marked by the red dashed line. A scatter plot of the raw data underlying each correlation is also shown in (b).

Comment 4 and 5

The authors expanded the discussion as requested.

Reading the answer caused me to speculate that the effect of the VSP70 allele on translational fidelity might be an artifact of the luciferase assay. The method does not measure the error rate directly, but rather the concentration of erroneously translated firefly luciferase, which can potentially be affected by protein sorting and degradation. Could the authors comment on this?

Response:

Thank you. We agree with you, and have now explicitly clarified this in the Result section when we explain the dual luciferase system. This is pasted below.

Note that the translation error rate measures both the effect of synthetic error and the sorting/degradation of erroneous proteins.

Minor Comments

All minor comments have been addressed.

Response:

Thank you.

Point-by-point response to the comments

Reviewer #6:

Comment 3

Thank you for the analyses. I'm not sure what short-living segregants were used in them. My question was specifically about whether the correlation between lifespan and error rate will be visible for segregants having the BY alleles on ChrXII:660,371 and on ChrXIV:461,485. However, I admit that I might not have explained it sufficiently, so I don't expect any additional analyses here.

I think the issue should be discussed directly in the manuscript fragments that refer to the effects of the VPS70 alleles.

Response:

Thank you. In response to your question, we tried correlation tests on segregants having the BY alleles on ChrXII:660,371 and/or on ChrXIV:461,485. In agreement with our model and your prediction, we found that the fidelity-longevity correlation never reaches statistical significance. These has now been added to **Supplementary Figure 7g-i**, and explicitly mentioned in **Discussion**, as pasted below.

... On the contrary, the fidelity-longevity correlation is absent among strains with the other alleles on either or both strains (Supplementary Figure 7g-i). ...

Supplementary Figure 7. (g-i) There was no correlation between

chronological lifespan and translation error rate in segregants with the BY allele at ChrXII:660,371 (**g**), ChrXIV:461,485 (**h**), or both loci (**i**). The Spearman's Correlation Coefficient and corresponding *P* value are provided.